# Can Simple Averaging Defeat Modern Watermarks?

**Pei Yang**[*]   **Hai Ci**[*]   **Yiren Song**   **Mike Zheng Shou**[†]
Show Lab, National University of Singapore
yangpei@u.nus.edu, cihai03@gmail.com
yiren@nus.edu.sg, mike.zheng.shou@gmail.com

## Abstract

Digital watermarking techniques are crucial for copyright protection and source identification of images, especially in the era of generative AI models. However, many existing watermarking methods, particularly content-agnostic approaches that embed fixed patterns regardless of image content, are vulnerable to steganalysis attacks that can extract and remove the watermark with minimal perceptual distortion. In this work, we categorise watermarking algorithms into content-adaptive and content-agnostic ones, and demonstrate how averaging a collection of watermarked images could reveal the underlying watermark pattern. We then leverage this extracted pattern for effective watermark removal under both greybox and blackbox settings, even when the collection of images contains multiple watermark patterns. For some algorithms like Tree-Ring watermarks, the extracted pattern can also forge convincing watermarks on clean images. Our quantitative and qualitative evaluations across twelve watermarking methods highlight the threat posed by steganalysis to content-agnostic watermarks and the importance of designing watermarking techniques resilient to such analytical attacks. We propose security guidelines calling for using content-adaptive watermarking strategies and performing security evaluation against steganalysis. We also suggest multi-key assignments as potential mitigations against steganalysis vulnerabilities. Github page: `https://github.com/showlab/watermark-steganalysis`.

## 1 Introduction

Digital watermarking hides information within digital media, facilitating copyright protection and source authentication [1–3]. With the advances in AI-based image generation and editing [4–7], robust and secure digital watermarking is crucial for preventing deepfake misuse or manipulations of created contents [8, 9].

We categorise digital watermarking methods into two types: content-adaptive and content-agnostic. Content-adaptive methods take images into the watermarking process, dynamically adjusting the watermark's placement and strength based on the image content, as seen in technologies like HiD-DeN [10] and RivaGAN [11]. Content-agnostic methods, however, use fixed, predefined watermark patterns independent of or weakly dependent on image content. Apart from traditional methods like DwtDctSvd [12], this also includes RoSteALS [13] that adds image-independent additive perturbations, and Tree-Ring [14] that places a ring pattern to the initial noise of a diffusion generation process. Content-adaptive methods typically offer better robustness against image processing distortions, while content-agnostic methods are computationally lighter and easier to implement.

A fundamental requirement for digital watermarks is robustness, ensuring watermarks cannot be easily removed or tampered with [2]. To meet the requirement, existing methods have been improving

---

[*]Equal contribution.
[†]Corresponding author.

watermark robustness through design considerations [14] or data augmentation during training [10], and have demonstrated strong robustness to various image distortions like noise perturbations or JPEG compression [15, 14, 16]. Works like Tree-Ring [14] even demonstrated robustness against strong attacks like VAE compression and image re-generation [17, 18]. In this paper, however, we reveal that content-agnostic watermarking techniques, including Tree-Ring, are vulnerable to steganalysis attacks, unmasking their hidden fragility.

To the best of our knowledge, our steganalysis is the first successful blackbox attack against Tree-Ring watermarks [14]. We discovered a content-agnostic ripple pattern in Tree-Ring-watermarked images and identified that this component is essential for watermark detection. Subtracting this pattern allows evading watermark detection with minimal impact on image perceptual quality. This raises the question: Do diffusion model watermarking methods that modify initial noise [19, 20, 14, 16] truly add semantic watermarks, or do they merely propagate low-level content-agnostic patterns to generated images?

Lastly, we propose new security guidelines for the watermarking community, emphasizing the importance of robustness against steganalysis. The guidelines call for performing evaluations against steganalysis when proposing new watermarking methods. It also encourages the development of content-adaptive watermarking methods to enhance resistance to steganalysis. For existing content-agnostic watermarking methods, we suggest assigning multiple watermarks per user as a mitigation strategy. In summary, the main contributions of this paper are:

- We reveal the vulnerability of content-agnostic watermarking methods to steganalysis removal and forgery.

- To the best of our knowledge, we are the first to successfully attack Tree-Ring watermarks in a blackbox setting, providing deeper insights into the essence of diffusion noise watermarking methods.

- We propose new security guidelines for future watermarking methods to help defend against steganalysis attacks.

## 2   Related works

### 2.1   Digital image watermarking

The field of digital image watermarking has evolved from traditional rule-based approaches to more recent deep learning-based techniques, with a significant focus on watermarking diffusion-generated images [14, 21, 22]. We categorise digital watermarking technologies into content-agnostic, which craft modifications based solely on the watermark information, and content-adaptive, which tailor modifications based on both watermark information and the image content.

**Content-agnostic watermarking**   Traditional methods like DwtDctSvd [12] employ fixed watermark patterns in transform domains, while more recent approaches modify the initial noise for diffusion-based image generation. Tree-Ring watermarks [14] replace the low-frequency Fourier-domain pixels of Gaussian noise with a ring pattern before using it for diffusion denoising. Similarly, Gaussian Shading [19] preserves the distribution while sampling the initial noise. Other approaches [13, 23, 24] train encoders to generate additive watermark perturbations without conditioning on image features.

**Content-adaptive watermarking**   These techniques leverage image features to generate watermarks tailored to the input image content. Early encoder-decoder methods like HiDDeN [10] and StegaStamp [15] employed deep neural networks to imprint watermarks onto the images. SSL [25] leveraged self-supervised networks as feature extractors, while RivaGAN [11] used attention mechanisms to look for appropriate local regions for watermark encoding. Recent approaches like Stable Signature [22], WADiff [21], and Zhao et al. [26] finetune the diffusion model to enable content-aware watermarking of diffusion-generated images. WMAdapter [27] designs a dedicated contextual adapter.

Through the classification, we highlight the vulnerability of content-agnostic techniques to steganalysis attacks, as they employ fixed or weakly content-dependent watermark patterns.

## 2.2 Attacks on watermarking

Traditional attacks applied distortions to disrupt watermarks, performing signal-level distortions like image compression, noise perturbation, blurring or colour adjustment, and geometric transformations like rotation or cropping [28–30, 10, 12]. In terms of videos, codecs may also distort invisible watermarks [31, 32]. These attacks fool watermark detectors at the cost of significant image quality degradation. To resist such attacks, training-based methods have been simulating the distortions via "attack layers" during training [10, 33–35], while training-free methods have been employing design considerations such as watermarking only low-frequency components [14].

Recent attacks base on deep models: regeneration attacks with diffusion models [36] and VAEs can provably remove pixel-level invisible watermarks [37, 18]. But such attacks are shown to be ineffective [17] for Tree-Ring [14] that alters the image a lot. When attackers can access the watermarking algorithm, they may also perform adversarial attacks [38]. The downside is that both regeneration and adversarial attacks are computationally expensive. In contrast, we propose a new type of blackbox steganalysis attack, which is efficient and works for content-agnostic watermarks. Steganalysis can extract meaningful watermark patterns, thus promoting further applications like forgery or explainability.

# 3 Watermark steganalysis

## 3.1 Notations

Let $x_\varnothing$ denote the original digital image, $w$ the watermark information (e.g., bit sequence or geometric pattern), and $E$ the watermark encoder that imprints $w$ into $x_\varnothing$, yielding the watermarked image $x_w = E(x_\varnothing, w)$. The embedding constraint ensures that $x_\varnothing$ and $x_w$ are perceptually indistinguishable. A watermark decoder $D$ recovers the embedded information $\hat{w} = D(x_w)$ for authentication purposes.

## 3.2 Threat model

The adversary aims to fool $D$ by manipulating $x_w$ using a strategy denoted as $T(\cdot)$ such that $D(T(x_w)) \neq w$ (watermark removal) or manipulating $x_\varnothing$ such that $D(T(x_\varnothing)) = w$ (watermark forgery). Formally, the adversary solves:

$$
\begin{aligned}
\text{Watermark Removal:} \quad & \max_T \|D(T(x_w)) - w\|, \\
\text{Watermark Forgery:} \quad & \min_T \|D(T(x_\varnothing)) - w\|,
\end{aligned}
\tag{1}
$$

subject to the constraint that the original image $x$ and the manipulated image $T(x)$ are perceptually indistinguishable. Rather than applying strong distortions as $T$, we demonstrate that the adversary can take a steganalysis approach to fool $D$.

## 3.3 Steganalysis: watermark extraction, removal and forgery

Figure 1 illustrates our watermark removal/forgery strategy $T$, which assumes that $E$ perturbs an additive pattern $\delta_w$ agnostic to image content, such that $x_w = x_\varnothing + \delta_w$. This assumption can be refined based on a detailed understanding of specific watermarking algorithms (as will be showcased in Section 4.4.1). Under this additive assumption, to either remove or forge watermarks, we can approximate $\hat{\delta}_w = x_w - x_\varnothing$. To improve approximation and reduce randomness, we propose averaging over $n$ images during pattern extraction:

$$
\hat{\delta}_w = \frac{1}{n} \left( \sum_{i=1}^n x_{w,i} - \sum_{i=1}^n x_{\varnothing,i} \right).
\tag{2}
$$

With the approximated $\hat{\delta}_w$, the adversary can perform greybox watermark removal ($\hat{x}_\varnothing = T(x_w) = x_w - \hat{\delta}_w$) or forgery ($\hat{x}_w = T(x_\varnothing) = x_\varnothing + \hat{\delta}_w$) on a given image $x$. Even without paired $x_\varnothing$, the adversary can perform blackbox removal/forgery by approximating $x_\varnothing$ through averaging any

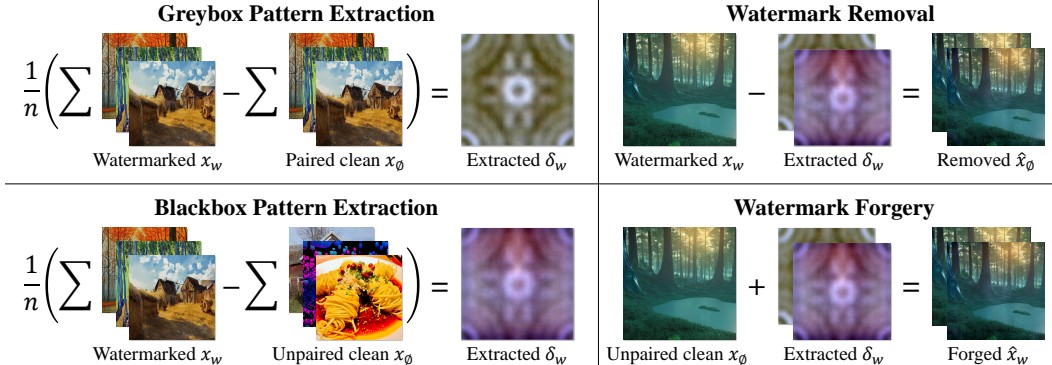

Figure 1: Watermark pattern extraction, removal and forgery under the Simple Linear Assumption. Two groups of paired (greybox) or unpaired (blackbox) images are first averaged and then subtracted to reveal the watermark pattern. The pattern extracted is then used for watermark removal/forgery.

collection of clean images from the Internet. There is a practical scenario where the adversary's watermarked image collection contains multiple different watermarks, we show in Section 4.4.4 that they can still use Equation 2 for pattern extraction.

# 4 Experiments

## 4.1 Experimental setup

**Image experiments** We evaluate our proposed steganalysis on 12 existing image watermarking methods: Tree-Ring [14], RingID [16], RAWatermark [23], DwtDctSvd [12], RoSteALS [13], Gaussian Shading [19], Stable Signature [22], WmAdapter[27], RivaGAN [11], SSL [25], HiDDeN [10], and DwtDct [12]. For the greybox setting, we use the COCO2017 [39] validation set for WmAdapter [27] and Stable Signature [22], Stable Diffusion Prompts [40] for Tree-Ring [14] and RingID [16] prompts, and DiffusionDB [41] for the remaining methods as the non-watermarked images ($x_\varnothing$). The corresponding watermarked images ($x_w$) are generated using the respective watermarking methods. In the blackbox setting with no access to paired images, we substitute $x_\varnothing$ with ImageNet [42] test set. The selection of images within the datasets is random. The datasets are resized to 256×256 for RoSteALS, SSL, and HiDDeN, and 512×512 for other methods.

We assess watermark removal under different $n$ (number of images averaged) during watermark pattern extraction, and test on 100 images[3] during watermark removal. We report detection AUC for Tree-Ring [14] and RAWatermark [23], and watermark decoding bit accuracy for the other methods. Additionally, we evaluate the image quality between $x_w$ and its non-watermarked counterpart, reporting PSNR in the main text and SSIM, LPIPS [43], and SIFID [44] in the appendix.

**Audio experiments** We then extend the experiments to audio watermark removal on AudioSeal [45] and WavMark [46], using the zh-CN subset of the Common Voice dataset [47]. Each audio segment is preprocessed to a 16 kHz mono format, with only the first two seconds retained. We use paired audio for greybox removal, and unpaired audio for blackbox removal. We report the watermark detection accuracy for AudioSeal [45], and watermark decoding bit accuracy for WavMark [46]. To quantify the audio quality after watermark removal, we calculate the Scale-Invariant Signal-to-Noise Ratio (SI-SNR) between the watermark-removed audio and its non-watermarked counterpart.

**Computing resources** The experiments were conducted on an AMD EPYC 7413 24-Core Processor and an Nvidia RTX 3090 GPU, requiring around 200GB of disk space. The execution time for each experiment ranges from around 10 minutes (HiDDeN) to around 10 hours (Tree-Ring).

---

[3]Each configuration is tested on 100 images or audio segments to minimise computational cost during repeated ablation studies on watermark removal and forgery.

## 4.2 Quantitative analysis on watermark removal

As shown in Figure 2 (left column), our steganalysis-based watermark removal method effectively degrades the detection performance of RAWatermark (0.5744 AUC), DwtDctSvd (0.5722 accuracy), Tree-Ring (0.2407 AUC), RoSteALS (0.2444 bit accuracy), and Gaussian Shading (0.5615 bit accuracy). The results highlight two key findings: (1) the aforementioned methods embed content-agnostic watermarks, and (2) content-agnostic watermarks are susceptible to steganalysis-based removal.

The effectiveness of our method increases as $n$ decreases, albeit at the cost of increased image distortion (smaller PSNR). In contrast, content-adaptive watermarking methods (Figure 2 right column) demonstrate robust resistance to this attack, maintaining high detection accuracy ($> 0.95$) upon convergence. This resilience underscores the importance of content-adaptivity in watermark design to thwart steganalysis-based removal attacks.

## 4.3 Qualitative analysis

In this qualitative analysis, we first examine the patterns extracted from various watermarking methods, then discuss how the removal of these watermarks affects image quality.

**Extracted patterns**    Figure 9 displays patterns extracted from content-agnostic methods, while Figure 10 shows those from content-adaptive methods. Content-agnostic watermarks tend to exhibit unique patterns, like fingerprints. For example, RingID [16] patterns are concentric rings with bright spots in the centre, DwtDctSvd [12] patterns resemble vertical lines like barcodes, and RoSteALS [13] patterns appear as grid-like patches with non-uniform illumination. In contrast, patterns extracted from content-adaptive watermarks are less discernible. Notably, under the greybox setting, the HiDDeN-extracted [10] pattern converges to zero, indicating completely no discernible pattern. Furthermore, patterns extracted in the greybox setting contain fewer visual artifacts than those in the blackbox setting. As more images are averaged, the extracted watermark pattern becomes clearer and more precise, with fewer residual artefacts from image contents. A more detailed analysis is presented in Appendix A.3.

**Visual quality degradation**    Figures 12 and 13 show the visual impact of removing content-agnostic and content-adaptive watermarks, respectively. For all methods but Gaussian Shading [19], under the greybox setting, when more than 50 images are averaged, virtually no visual artefacts remain after watermark removal. In the blackbox setting, averaging over 100 images is necessary to eliminate most artifacts. The exception is Gaussian Shading [19], which consistently produces visible artefacts because the pattern extracted has a large magnitude. Subtracting this pattern significantly distorts the image. Appendix A.4 provides a more detailed analysis.

## 4.4 Case study: Tree-Ring watermarks

To further reveal how steganalysis can be a threat to content-agnostic watermarking algorithms, we conduct a case study on Tree-Ring watermarks [14]. Tree-Ring is a sophisticated diffusion-based watermarking algorithm that injects a frequency-domain ring pattern into a Gaussian noise signal before using this watermarked noise for diffusion-denoising image generation. During detection, it performs DDIM inversion to recover the injected ring pattern from the initial noise and compares it to a reference pattern. In the following experiments, we demonstrate that with minimal modifications, we can both remove and forge Tree-Ring watermarks under different scenarios.

### 4.4.1 Low-level content-agnostic pattern in Tree-Ring

This section focuses on revealing the low-level content-agnostic component of Tree-Ring watermarks [14]. First, we curate a specific steganalysis for Tree-Ring's detection algorithm, demonstrating how tailored steganalysis more accurately extracts watermark patterns than generic averaging. We then compare the extracted watermarks with the ground truth to showcase this low-level component.

Our steganalysis incorporates the DDIM inversion steps from Tree-Ring's detection process. By inverting watermarked images to the DDIM-inverted latent space and averaging them, we obtain patterns (Figure 3, second row). As more images are averaged, these patterns closely resemble those

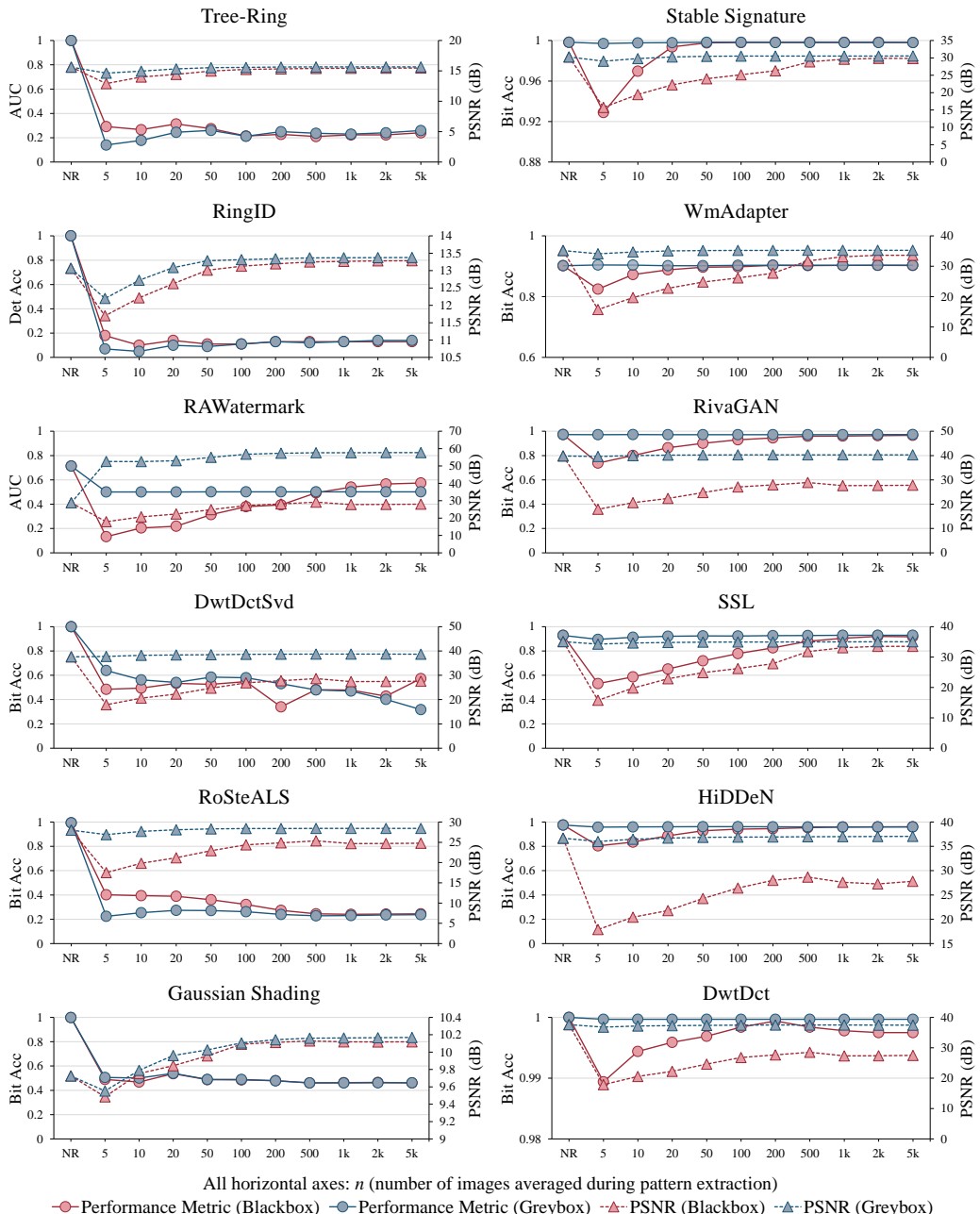

Figure 2: Performance of watermark detectors under steganalysis-based removal. Performance metrics include AUC (watermark verification AUC), Bit Acc (bit accuracy, percentage of correctly decoded bits), and Det Acc (detection accuracy, accuracy of fully recovered watermark). The plots also illustrate the corresponding PSNR as a measure of image quality degradation. The left/right columns show content-agnostic/content-adaptive methods, respectively. NR denotes the case without removal, reflecting the decoder's inherent performance.

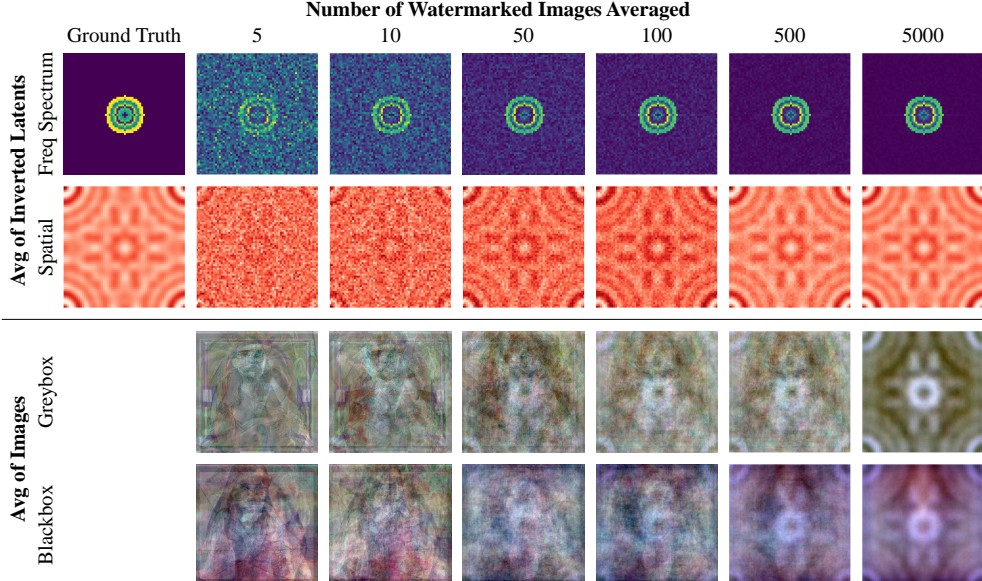

**Figure 3:** Visualisation of Tree-Ring-extracted watermark patterns. Top: Pattern extracted from DDIM-inverted latents without subtracting $x_\varnothing$. The first and second rows are Fourier transform pairs. Bottom: Pattern extracted in image space under greybox and blackbox settings, akin to Figure 9.

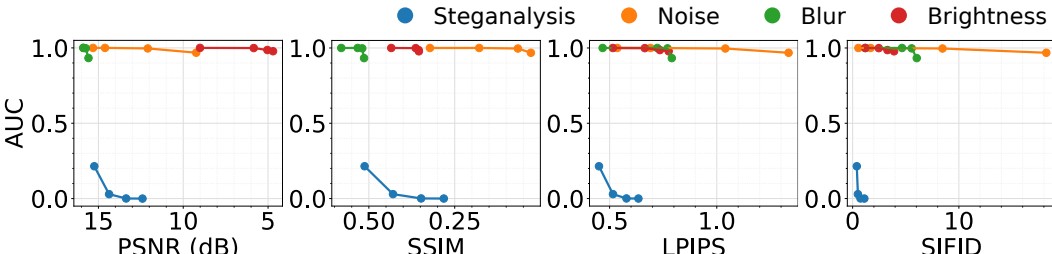

**Figure 4:** Tree-Ring detection AUC against quality metrics for steganalysis-based removals ($n = 5000$) and image distortions. Steganalysis-based removals (blue) cluster bottom-left, indicating effective watermark removal with comparatively low quality degradation.

extracted under greybox or blackbox settings, manifesting as ripples spreading from the corners and forming aliasing patterns in the centre, reminiscent of superpositioned 2D sinc functions.

In the Fourier domain (Figure 3, first row), these patterns display a clear ring structure nearly identical to the ground truth. The high similarity between the ground truth and the patterns extracted from both the image and DDIM-inverted latent domains indicates that **Tree-Ring likely propagates a content-agnostic ripple pattern throughout the image generation process, slightly but directly revealing it in the generated images**. This insight enables us to fool Tree-Ring's watermark detector by simply subtracting this ripple pattern, effectively removing the watermark information.

### 4.4.2 Comparison with distortion-based removal techniques

To compare perceptual quality degradation between our method and distortion-based ones, in Figure 4, we plot Tree-Ring's AUC versus qualitative metrics varying signal strengths during watermark subtraction. In all four plots, the steganalysis-based watermark removal curves clustered in the bottom-left corner, indicating that effective steganalysis can remove watermarks with significantly less image quality degradation compared to distortion-based methods. Note that although in Section 4.2, the excess performance degradation under small $n$ is believed to be caused by excess distortions

Table 1: Tree-Ring [14] detection accuracy at 1% FPR for watermark removal and forgery. NRmv represents "no removal".

| # Imgs Avged | NRmv | 5 | 10 | 20 | 100 | 200 | 500 | 1000 | 2000 | 5000 |
|---|---|---|---|---|---|---|---|---|---|---|
| **Removal** | 1.00 | 0.08 | 0.09 | 0.13 | 0.13 | 0.13 | 0.14 | 0.14 | 0.14 | 0.14 |
| **Forgery** | 1.00 | 0.00 | 0.00 | 0.00 | 0.00 | 0.00 | 0.00 | 0.00 | 0.00 | 0.00 |

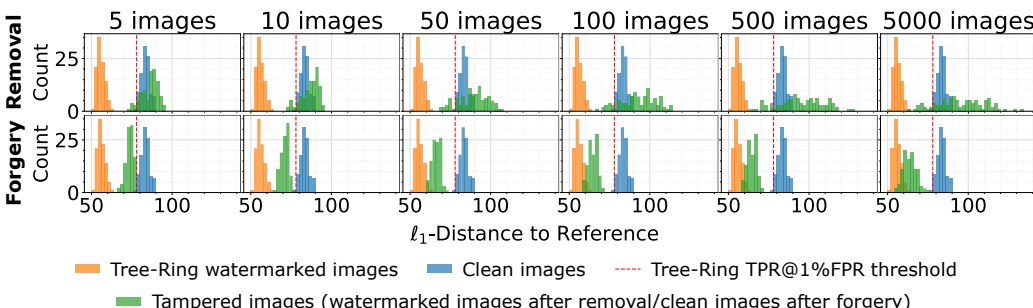

Figure 5: Histograms of distance to reference watermarking pattern for Tree-Ring watermark removal (top) and forgery (bottom). For removal, averaging more images pushes the watermark-removed images (green) away from the true watermarked images (orange). For forgery, oppositely, this increases the similarity of forged images (green) to true watermarked images (orange). Red dashed lines are thresholds $\tau$ at 1% FPR.

introduced due to imperfect pattern extraction, in this section, we demonstrate that distortions generally do not help remove watermarks. Appendix A.2 visualises images under these distortions.

### 4.4.3 Watermark forgery

We demonstrate the ability to forge Tree-Ring watermarks ($\hat{x}_w = x_\varnothing + \hat{\delta}_w$) on non-watermarked images, in addition to watermark removal ($\hat{x}_\varnothing = x_w - \hat{\delta}_w$). Table 1 shows the forged watermarks completely deceive Tree-Ring's detection. Figure 5 shows forged watermarks exhibit slightly larger distances compared to authentic watermarked images. However, when $n$ is large (500 images), the forged images overlap with true watermarked images in the histogram, precluding threshold-based separation, thereby demonstrating Tree-Ring's vulnerability to steganalysis-based watermark forgery.

### 4.4.4 Effectiveness of removal under multiple watermarks

We study a heterogeneous scenario where the adversary's image collection contains multiple different watermark patterns. When there are more patterns in the adversary's image collection, the detection AUC rises while the PSNR drops, indicating decreased steganalysis removal efficacy. Mixing three different watermark patterns increases Tree-Ring's detection AUC from below 0.2 to above 0.7 in both greybox and blackbox settings with $n = 5000$, indicating that mixing watermarking keys could

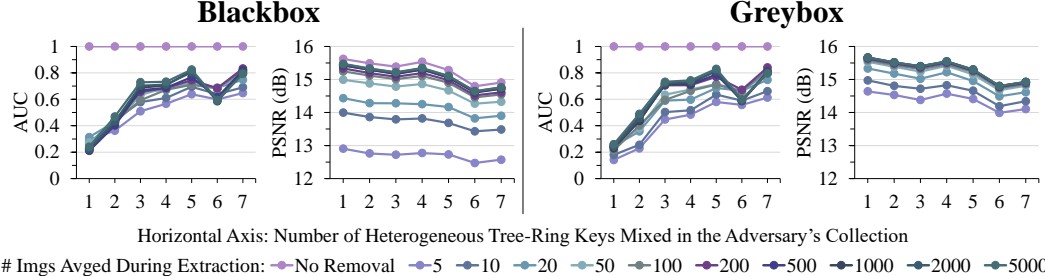

Figure 6: Ablation study on watermark removal performance with Tree-Ring [14] when the adversary's image collection contains multiple watermark patterns.

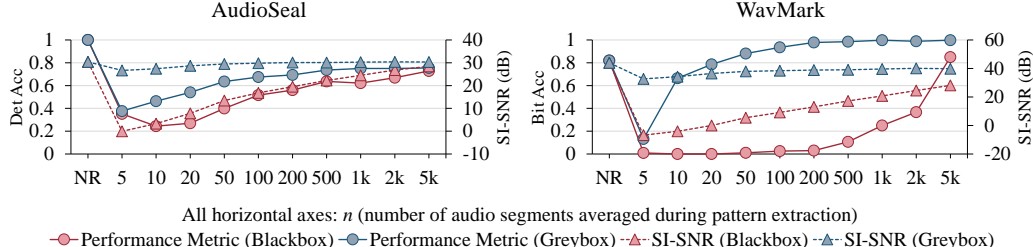

Figure 7: Impact of steganalysis-based removal on audio watermark detection. The plot shows watermark detection accuracy (Det Acc) for AudioSeal [45] and bit accuracy (Bit Acc) for WavMark [46]. SI-SNR values indicate audio quality changes post-removal. NR represents the baseline performance without any removal attempts.

improve security against simple steganalysis-based watermark removals. Nevertheless, the remaining 0.3 gap to perfectness still demonstrates the vulnerability of content-agnostic watermarking. We highlight that assigning multiple watermarks serves as a mitigation and cannot fundamentally address the steganalysis vulnerability. (Appendix A.6 gives more cases).

### 4.4.5 Summary

In this case study, we rooted Tree-Ring's security vulnerabilities in its use of low-level content-agnostic ripple patterns as watermarks, rather than solely from semantic watermarking. This enables us to successfully fool Tree-Ring watermark detection with minimal impact on perceptual quality. Although it exhibits strong robustness to distortions [14] and regeneration attacks [18, 17], through steganalysis-based removal, we are the first to effectively remove Tree-Ring watermarks without access to the algorithm.

### 4.5 Audio watermark steganalysis

The distinction between content-agnostic and content-adaptive watermarks extends beyond images, applying equally to other media like audio. To test the generality of our steganalysis approach, we extend the experiments to two audio watermarking methods: AudioSeal [45] and WavMark [46].

Following the methodology outlined in Section 3.3, we extract audio watermark patterns by averaging in the time domain. Figure 7 illustrates the efficacy of this approach on audio watermarks. Similar to content-agnostic image watermarks, our steganalysis-based removal significantly impairs the performance of AudioSeal, reducing its detection accuracy from a perfect 1.0 to around 0.75 in both greybox and blackbox settings. This decline underscores AudioSeals's vulnerability to our simple averaging-based steganalysis.

Interestingly, for WavMark, subtracting the averaged pattern counterintuitively improves its bet accuracy from below 0.8 to a perfect 1.0 when $n$ is large. While the complexity of WavMark's algorithm precludes definitive conclusions from this experiment, the pattern extracted under large $n$ showcased the existence of the systematic bias and its correlation with the watermark information. Although our method does not directly "remove" WavMarks' watermark in the traditional sense, the observed behaviour raises questions about its resilience to more sophisticated steganalysis attacks.

In both cases, smaller $n$ values lead to lower watermark detection rates and lower SI-SNR values, demonstrating coarsely extracted patterns further degrading detection performance with additional audio quality distortions. This mirrors our finding in the image domain, highlighting the importance of sufficient large $n$ for more accurate watermark pattern extraction across different media types.

## 5   Guidelines towards steganalysis-secure watermarking

Our analysis demonstrates that content-agnostic watermarking methods like Tree-Ring [14] are vulnerable to steganalysis-based attacks. Although these methods claim robustness by demonstrating strong resistance against distortions (e.g., blurring or noise perturbations), adversaries may still

remove the watermark through steganalysis, thus compromising their robustness. Our experiments reveal that even complex, highly nonlinear methods based on deep neural networks are susceptible to steganalysis-based watermark removal.

To avoid such threat, future watermarking algorithms shall use content-adaptive watermarking methods. One approach is to incorporate image features while encoding watermark information. For example, HiDDeN [10] and RivaGAN [11] introduce image features into the watermark encoder through concatenation and attention, respectively. For existing content-agnostic methods, assigning multiple watermarks per user can *mitigate* but not *fundamentally* solve steganalysis threat.

While we performed simple averaging steganalysis on RGB images and 16 kHz monophonic audios, watermarking methods should resist more complex techniques, such as steganalysis in various colour spaces or different transform domains. Evaluating security against diverse steganalysis models, analogous to robustness tests against distortions, is crucial for developing secure watermarking algorithms. These two aspects form our security guidelines:

1. Ensure watermarks are content-adaptive to resist steganalysis attacks.
2. Evaluate watermark against steganalysis to ensure robustness.

## 6 Conclusions

This work has revealed the vulnerability of content-agnostic watermarking algorithms to steganalysis attacks. We have demonstrated effective watermark removal and forgery techniques under both greybox and blackbox settings across twelve watermarking methods, including recent deep-learning approaches. Our findings extend to audio watermarking methods as well. To address these threats, we propose security guidelines that encourage exploring content-adaptive watermarking methods and evaluating them against steganalysis attacks. We have also proposed temporary mitigations for existing content-agnostic methods. Only by addressing the vulnerability to steganalysis can we develop secure and robust digital watermarking systems capable of safeguarding the integrity of digital content in the era of generative AI.

**Limitations and ethical considerations**   Steganalysis-based watermark removal/forgery is only effective against content-agnostic watermarking, not content-adaptive techniques. Responsible development and deployment of steganalysis technologies are crucial, adhering to fairness, accountability, and transparency principles to prevent misuse for unwarranted surveillance or privacy violations.

**Broader impacts**   The proposed steganalysis attack and security guidelines extend beyond image watermarking. They apply to watermarking other media like video [48], audio [49], 3D models [50–52], and to other domains. Our proposed guidelines and mitigation strategies strengthen watermarking security, contributing to a safer digital environment.

## 7 Acknowledgements

This research is supported by the Ministry of Education, Singapore, under the Academic Research Fund Tier 1 (FY2022) Award 22-5406-A0001.

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

# A  Supplementary Materials

## A.1   Watermark implementation details

**Tree-Ring watermarks** [14] are implemented following the corrected version proposed by RingID [16] under the verification setting. In comparison, the original Tree-Ring watermarks exhibited an issue where the watermark pattern actually injected into the initial noise pattern was inconsistent with the pattern claimed by Tree-Ring [14].

Our implementation adopts the original ring pattern proposed by Tree-Ring [14] with radii from 0 to 10, but the ring pattern is centred at the origin in the Fourier domain, following the correction proposed by RingID [16]. To ensure lossless watermark injection, we discard the imaginary part of the sampled watermark pattern during watermark detection. This lossless injection ensures that the watermark pattern actually injected matches the reference pattern used during detection.

**RingID** [16] is implemented using patterns with radii ranging from 3 to 14 in the identification setting.

## A.2   Perceptual quality of images under distortions

The distortions applied in plotting Figure 4 are violent, as visualised in Figure 8. Blurring with a radius of 13 renders the cat's eyes invisible. Perturbing with $\sigma = 100$ noise eliminates the cat's pattern. Upscaling the brightness to eight times overexposes the entire background. However, as shown in Figure 4, none of these distortions can defeat Tree-Ring's detector, but steganalysis-based watermark removal can. This is possibly due to Tree-Ring's large-scale content-agnostic ripple-like

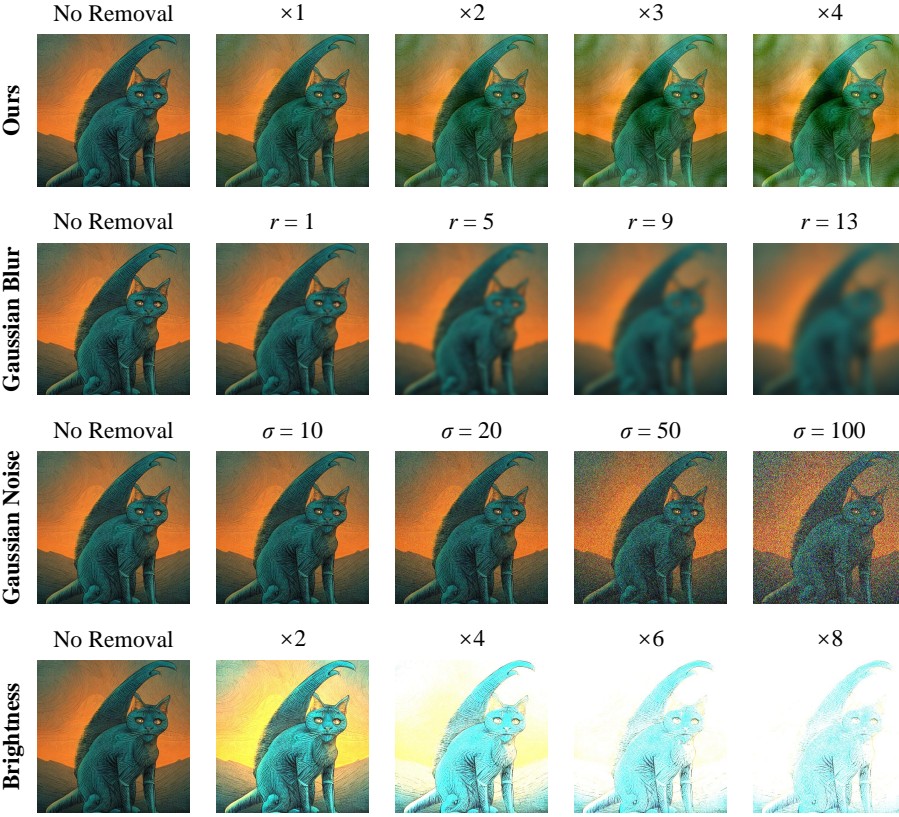

Figure 8: Visualisation of images under different strengths of steganalysis-based watermark removal (blackbox, $n = 5000$) and image distortions. We amplify the signal strength of the extracted pattern by multiplication with a factor.

patterns according to our analysis in the main paper. This demonstrates that steganalysis-based watermark removal effectively captures the vulnerability of the watermarking method through the extracted pattern, even without knowledge about the watermarking algorithm. Subsequently, it highlights that Tree-Ring's robustness to distortions is a *Maginot Line* that can be breached with steganalysis attacks.

## A.3 More visualisations of patterns extracted

In the main text, we have visualised the patterns extracted from content-agnostic watermarking methods. In this section, we present a more comprehensive visualisation of patterns extracted from content-agnostic methods in Figure 9, along with visualisations of the patterns extracted from content-adaptive watermarking techniques, as depicted in Figure 10.

While subtracting these patterns from watermarked images may not effectively circumvent the watermark detection process, the extracted patterns still exhibit regularities. By averaging 5000 images, the patterns extracted from both Stable Signature [22] and WmAdapter [27] display mild repetitive grid-like structures, suggesting that images watermarked with this technique may share commonalities at specific frequency components. For such repetitive patterns, it would be worth exploring whether they are caused by the patching mechanism of the transformer architecture. Patterns extracted by RivaGAN [11] exhibit a greenish tint, indicating RivaGAN introduces a systematic bias during the watermarking process despite content-adaptive components. The patterns extracted by SSL [25], similar to RAWatermark [23], are biased towards a specific noise distribution. However, while RAWatermark [23] is vulnerable to steganalysis-based removal, SSL [25] demonstrates robustness against such attacks. As the parameter $n$ increases, the noise components in the patterns extracted by HiDDeN [10] transition towards a greyish hue and eventually diminish. By the gray-world assumption principle, this observation suggests that HiDDeN best incorporates content-adaptive watermarks into the images. Under spatial domain averaging, DwtDct [12] reveals nothing but a tiny systematic bias, which aligns with the fact

We also visualise audio patterns extracted from AudioSeal [45] and WavMark [46] in Figure 11. When $n = 5000$, the greybox AudioSeal-extracted pattern is highly imperceptible, with a signal strength below -45dB relative to the cover media. However, directly subtracting this pattern in the time domain significantly reduces the watermark detection accuracy by 30%. In contrast, WavMark-extracted patterns have larger amplitudes, indicating that WavMark also introduces a systematic bias during the watermarking process. Moreover, greybox-extracted WavMark patterns concentrate within the first second, which aligns with WavMark's method of adding watermark patterns at one-second intervals. This exposes WavMark's watermarking locations, allowing an adversary to potentially remove the watermark by simply clipping out these segments. The spectrograms further show that WavMark mainly adds watermarks below 6 kHz, which could represent the robustness threshold it was trained to withstand against low-pass filtering. In summary, although both methods claim robustness against various distortion-based attacks, their vulnerabilities can be easily exposed through simple steganalysis techniques.

## A.4 Perceptual quality of images after watermark removal

In the main text, we analyzed the perceptual quality of Tree-Ring watermarked images after watermark removal. We now visualise content-agnostically watermarked images after watermark removal in Figure 12, and content-adaptively watermarked images after watermark removal in Figure 13.

From these two figures, we can observe that except for Gaussian Shading [19], images carrying different watermarks, after watermark removal, have nearly identical visual quality. The visual quality is predominantly related to $n$, the number of images averaged during pattern extraction. In the greybox setting, when $n > 50$, there are no visually apparent artefacts. In the blackbox setting, $n > 100$ is generally required for eliminating significant artefact patterns. For images with Gaussian Shading [19] watermarks, due to the large magnitude of the extracted watermark pattern itself, steganalysis-based removal inevitably causes significant changes to the image content, thereby reducing perceptual quality. This leads to two insights:

1. For adversaries, averaging more images allows for a better approximation of an effective content-agnostic watermark pattern that can be used for watermark removal;

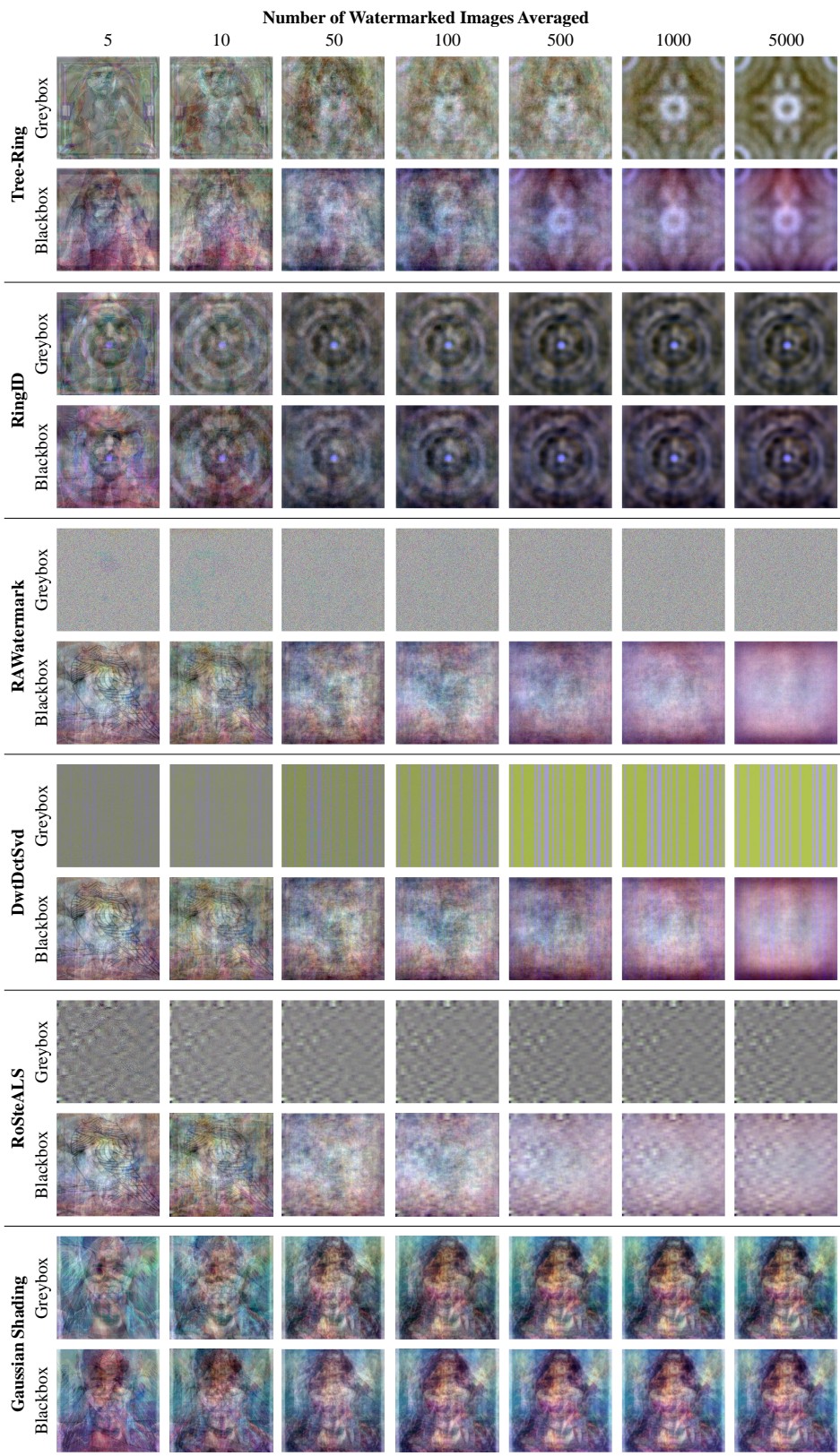

Figure 9: Visualisation of watermark pattern extracted from content-agnostic methods. All patterns are adaptively normalised before visualisation.

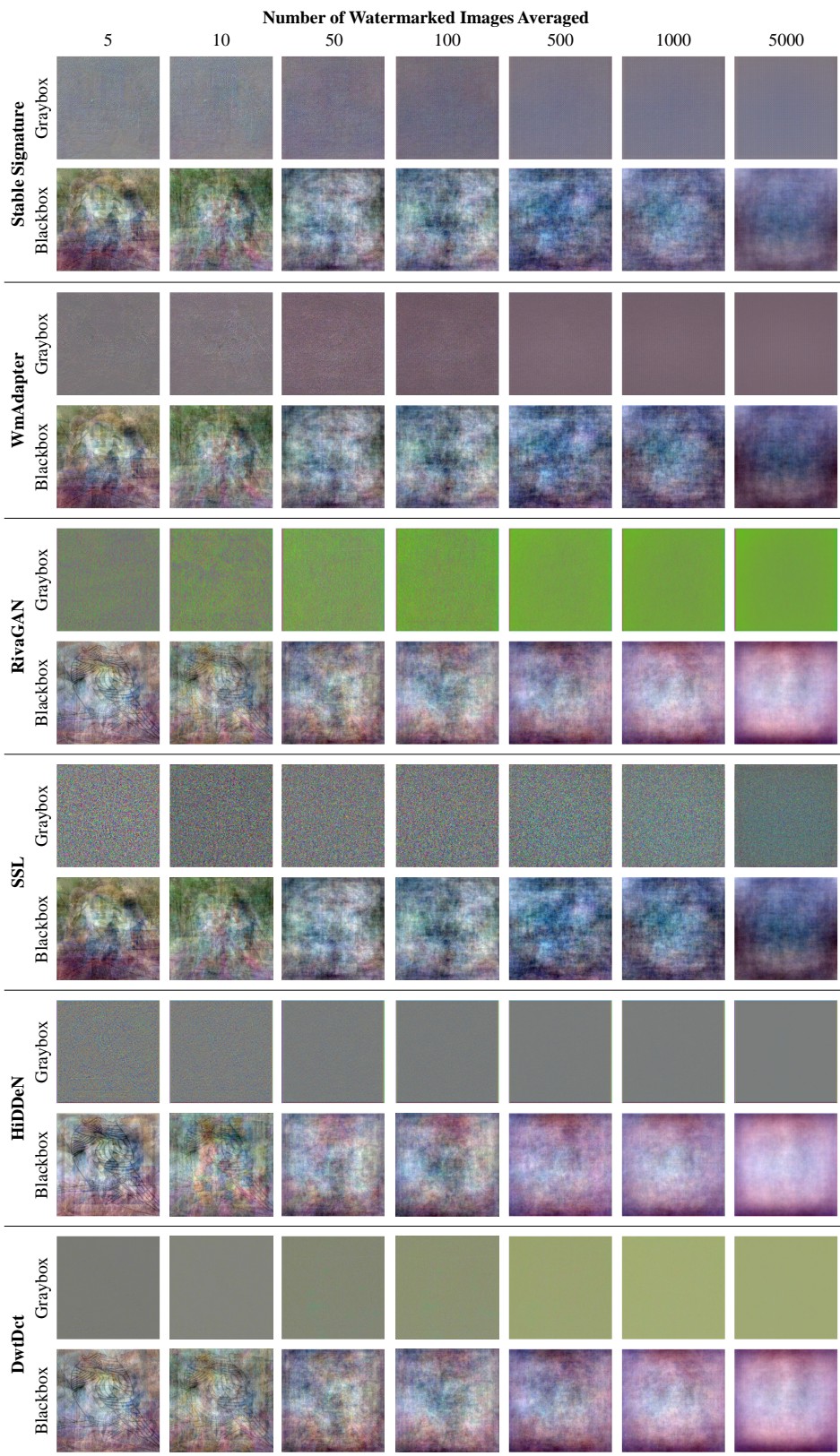

Figure 10: Visualisation of patterns extracted from content-adaptive watermarking methods. All patterns are adaptively normalised before visualisation.

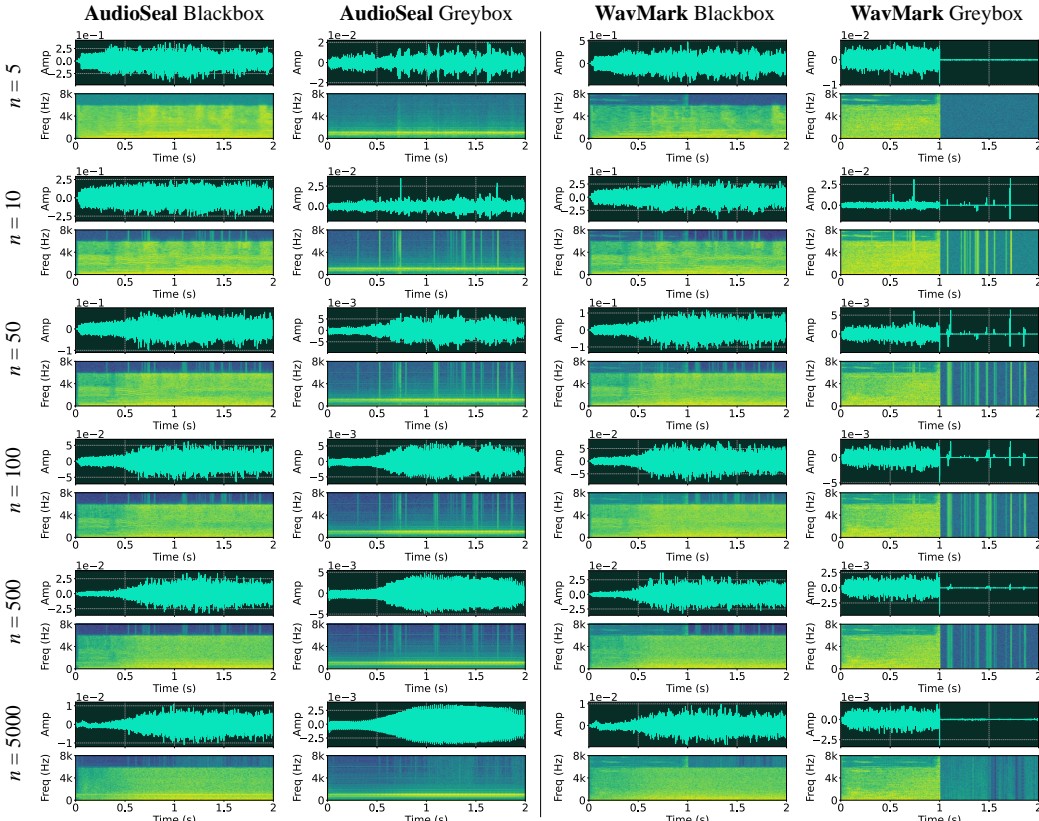

Figure 11: Visualisation of audio patterns extracted. Each subplot contains a time-domain audio signal (top) and its spectrogram (bottom). Amp stands for (time-domain) audio signal amplitude. The amplitudes are normalised to within [-1, 1].

2. For watermark distributors, reducing the count of distributing the same watermark could lower the security risk posed by steganalysis.

### A.5 More results on quantitative analysis on watermark removal

In Section 4.2, we primarily discussed the Peak Signal-to-Noise Ratio (PSNR) as a measure of image distortion resulting from our steganalysis-based watermark removal method. To provide a more comprehensive assessment of the impact on visual quality, we also evaluated three additional metrics: Structural Similarity Index (SSIM), Learned Perceptual Image Patch Similarity (LPIPS), and Single Image Fréchet Inception Distance (SIFID).

Tables 2-13 present the performance of various watermarking methods under our steganalysis-based removal attack, along with the corresponding image quality metrics. The trends observed in PSNR are largely mirrored in the additional metrics:

**Content-Agnostic Methods**    For methods like Tree-Ring [14], RingID [16], RAWatermark [23], DwtDctSvd [12], and RoSteALS [13], all metrics show a consistent trend: as the number of images averaged ($n$) decreases, watermark removal becomes more effective (lower detection rates), but at the cost of increased image distortion (lower PSNR, SSIM, and higher LPIPS, SIFID).

For example, in the case of RAWatermark (Table 4, Blackbox setting), as $n$ decreases from 5000 to 5, the AUC drops from 0.574 to 0.133, indicating more effective watermark removal. However, this comes at the cost of image quality: PSNR drops from 27.98 to 17.92, SSIM from 0.964 to 0.528, while LPIPS increases from 0.020 to 0.424, and SIFID from 0.022 to 1.401, all indicating significant visual degradation.

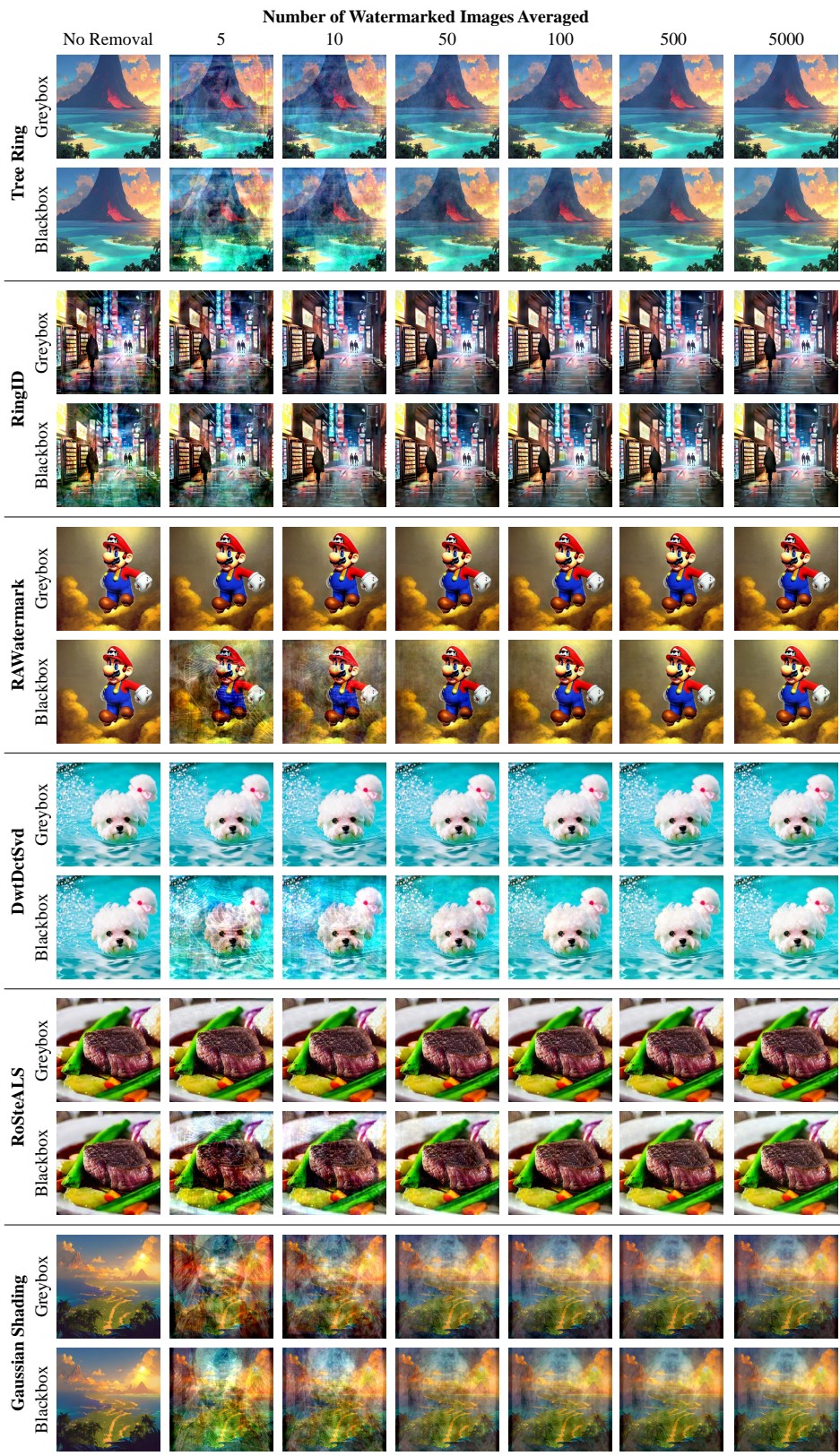

Figure 12: Visualisation of content-agnostically watermarked images after watermark removal.

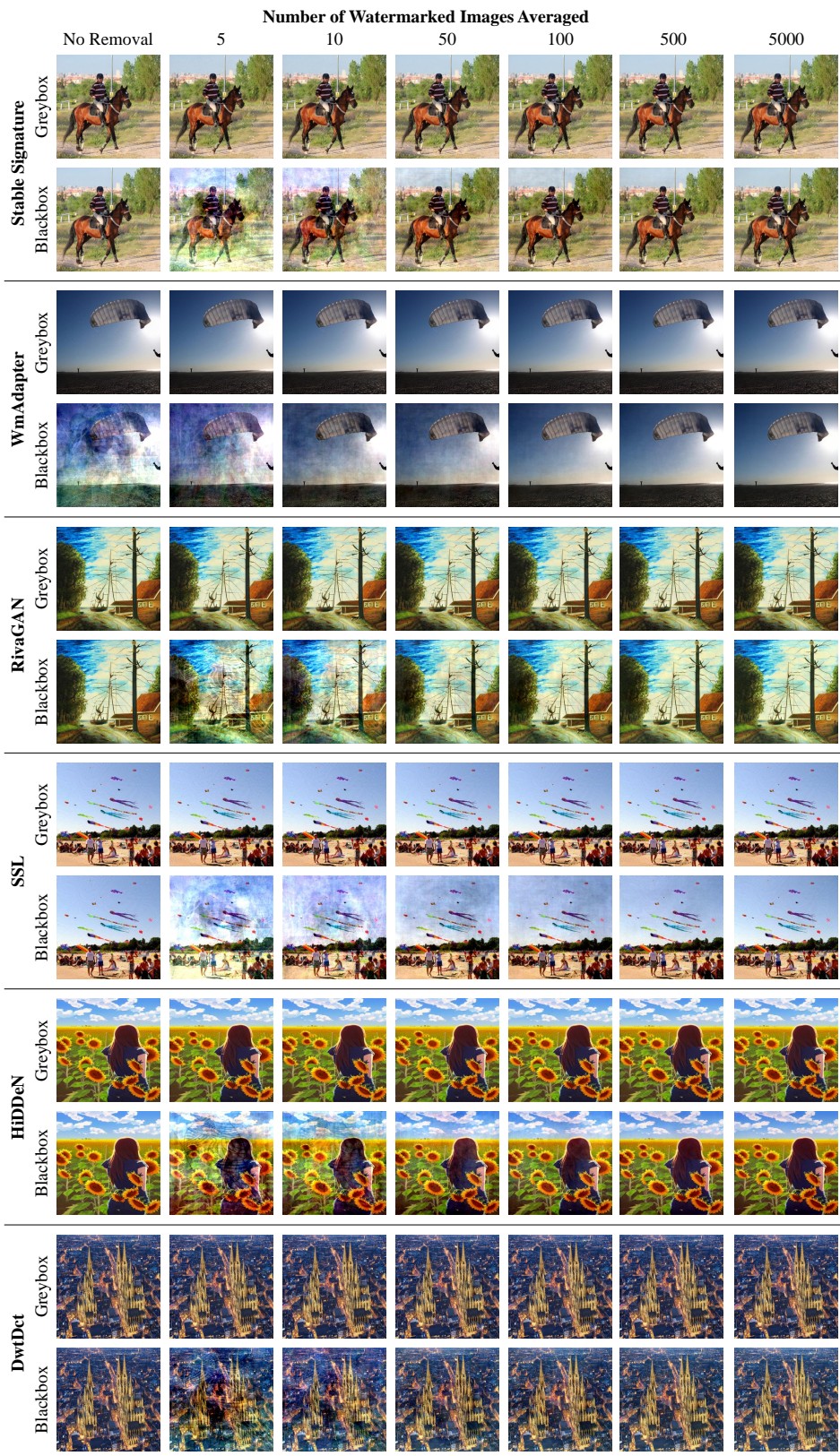

Figure 13: Visualisation of content-adaptively watermarked images after watermark removal.

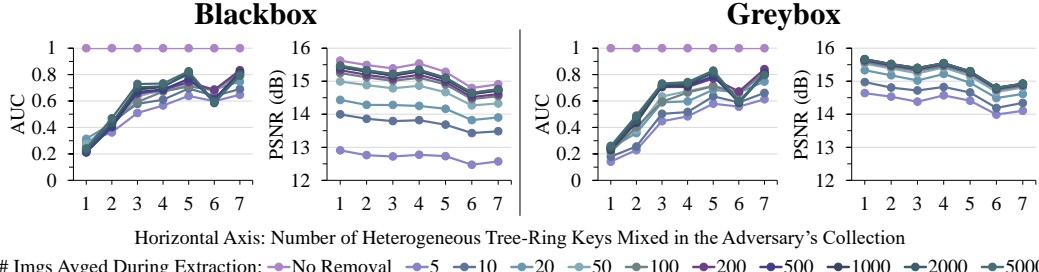

Figure 14: Ablation study on watermark removal performance with DwtDctSvd [12] when the adversary's image collection contains multiple watermark patterns.

**Content-Adaptive Methods**   In contrast, content-adaptive methods like Stable Signature [22], WmAdapter [27], RivaGAN [11], and SSL [25] maintain good detection performance (bit accuracies $> 0.95$) even as $n$ decreases, with the image quality metrics also show minimal degradation. For instance, SSL (Table 11, greybox setting) maintains a high bit accuracy (0.895 at $n = 5$) while preserving image quality: PSNR (34.26), SSIM (0.920), LPIPS (0.048), and SIFID (0.048) are all close to the no-removal (NRmv) values. This could be due to that the systematic biases averaged out from content-adaptive methods are mild, so subtracting such mild patterns from the original images would not lead to significant perturbations.

## A.6   Effectiveness of removal under multiple watermarks

In Section 4.4.4, our case study on Tree-Ring [14] demonstrated that when an adversary's image collection contains a mix of several different watermark patterns, the effectiveness of watermark removal is significantly reduced. Based on this finding, we proposed assigning multiple keys as a mitigation strategy against steganalysis threats. However, in this section, we caution that this approach is not a universal solution and should be applied judiciously. The high complexity of watermarking algorithms means that this method cannot guarantee enhanced watermark security. To illustrate this, we replicate our experiment using the DwtDctSvd [12] algorithm.

Figure 14 shows that in the greybox scenario, DwtDctSvd behaves similarly to Tree-Ring: as the adversary's image collection incorporates more diverse watermarks, watermark removal becomes less effective without significantly impacting image quality. Interestingly, this strategy fails in the blackbox setting. When the adversary's collection mixes more than 4 watermarks, DwtDctSvd's detection accuracy surprisingly drops below 0.1. While the reasons behind this phenomenon warrant further investigation, this observation underscores our main point: assigning multiple watermarks per user is merely a temporary workaround to bolster current content-agnostic watermarking algorithms. It does not address the fundamental vulnerabilities of these algorithms and may even be counterproductive in certain scenarios. Therefore, watermark distributors should employ this method cautiously and at

Table 2: Performance (AUC) of Tree-Ring [14] under steganalysis-based removal and the corresponding image quality degradations. NRmv stands for no removal.

| # Images Averaged | | NRmv | 5 | 10 | 20 | 50 | 100 | 200 | 500 | 1000 | 2000 | 5000 |
|---|---|---|---|---|---|---|---|---|---|---|---|---|
| | AUC | 1.000 | 0.293 | 0.267 | 0.314 | 0.275 | 0.214 | 0.228 | 0.211 | 0.224 | 0.224 | 0.241 |
| | PSNR | 15.62 | 12.91 | 14.00 | 14.43 | 14.99 | 15.24 | 15.33 | 15.42 | 15.43 | 15.46 | 15.47 |
| **Blackbox** | SSIM | 0.555 | 0.298 | 0.355 | 0.431 | 0.482 | 0.512 | 0.528 | 0.540 | 0.545 | 0.547 | 0.548 |
| | LPIPS | 0.411 | 0.609 | 0.566 | 0.514 | 0.472 | 0.451 | 0.440 | 0.430 | 0.427 | 0.425 | 0.425 |
| | SIFID | 0.375 | 1.205 | 0.751 | 0.461 | 0.409 | 0.392 | 0.396 | 0.402 | 0.404 | 0.404 | 0.404 |
| | AUC | 1.000 | 0.141 | 0.179 | 0.246 | 0.260 | 0.213 | 0.251 | 0.237 | 0.230 | 0.241 | 0.259 |
| | PSNR | 15.62 | 14.64 | 14.97 | 15.33 | 15.53 | 15.58 | 15.63 | 15.65 | 15.66 | 15.66 | 15.67 |
| **greybox** | SSIM | 0.555 | 0.346 | 0.401 | 0.473 | 0.511 | 0.531 | 0.542 | 0.549 | 0.551 | 0.552 | 0.553 |
| | LPIPS | 0.411 | 0.554 | 0.523 | 0.479 | 0.449 | 0.433 | 0.422 | 0.415 | 0.413 | 0.412 | 0.412 |
| | SIFID | 0.375 | 0.955 | 0.588 | 0.402 | 0.367 | 0.364 | 0.367 | 0.371 | 0.372 | 0.373 | 0.373 |

Table 3: Performance (decoding accuracy) of RingID [16] under steganalysis-based removal and the corresponding image quality degradations. NRmv stands for no removal.

| # Images Averaged | | NRmv | 5 | 10 | 20 | 50 | 100 | 200 | 500 | 1000 | 2000 | 5000 |
|---|---|---|---|---|---|---|---|---|---|---|---|---|
| | **Dec Acc** | 1.000 | 0.180 | 0.100 | 0.140 | 0.110 | 0.110 | 0.130 | 0.130 | 0.130 | 0.130 | 0.130 |
| | **PSNR** | 13.06 | 11.70 | 12.21 | 12.62 | 13.01 | 13.12 | 13.19 | 13.24 | 13.26 | 13.28 | 13.28 |
| **Blackbox** | **SSIM** | 0.459 | 0.240 | 0.276 | 0.345 | 0.392 | 0.418 | 0.430 | 0.441 | 0.446 | 0.448 | 0.449 |
| | **LPIPS** | 0.460 | 0.595 | 0.564 | 0.523 | 0.495 | 0.483 | 0.479 | 0.477 | 0.475 | 0.475 | 0.475 |
| | **SIFID** | 0.250 | 0.613 | 0.380 | 0.266 | 0.258 | 0.255 | 0.259 | 0.260 | 0.260 | 0.259 | 0.259 |
| | **Dec Acc** | 1.000 | 0.070 | 0.050 | 0.100 | 0.090 | 0.110 | 0.130 | 0.120 | 0.130 | 0.140 | 0.140 |
| | **PSNR** | 13.06 | 12.19 | 12.72 | 13.08 | 13.28 | 13.31 | 13.34 | 13.36 | 13.37 | 13.37 | 13.37 |
| **greybox** | **SSIM** | 0.459 | 0.245 | 0.293 | 0.368 | 0.410 | 0.430 | 0.440 | 0.446 | 0.449 | 0.450 | 0.451 |
| | **LPIPS** | 0.460 | 0.574 | 0.545 | 0.509 | 0.486 | 0.477 | 0.472 | 0.470 | 0.469 | 0.469 | 0.469 |
| | **SIFID** | 0.250 | 0.591 | 0.352 | 0.253 | 0.239 | 0.241 | 0.243 | 0.244 | 0.245 | 0.245 | 0.246 |

Table 4: Performance (AUC) of RAWatermark [23] under steganalysis-based removal and the corresponding image quality degradations. NRmv stands for no removal.

| # Images Averaged | | NRmv | 5 | 10 | 20 | 50 | 100 | 200 | 500 | 1000 | 2000 | 5000 |
|---|---|---|---|---|---|---|---|---|---|---|---|---|
| | **AUC** | 0.714 | 0.133 | 0.205 | 0.219 | 0.315 | 0.379 | 0.394 | 0.494 | 0.540 | 0.566 | 0.574 |
| | **PSNR** | 28.83 | 17.92 | 20.67 | 22.38 | 24.86 | 27.21 | 28.14 | 29.15 | 27.81 | 27.86 | 27.98 |
| **Blackbox** | **SSIM** | 0.928 | 0.528 | 0.659 | 0.764 | 0.862 | 0.916 | 0.939 | 0.960 | 0.959 | 0.961 | 0.964 |
| | **LPIPS** | 0.028 | 0.424 | 0.327 | 0.236 | 0.149 | 0.096 | 0.065 | 0.036 | 0.028 | 0.022 | 0.020 |
| | **SIFID** | 0.017 | 1.401 | 0.696 | 0.286 | 0.102 | 0.047 | 0.039 | 0.022 | 0.024 | 0.023 | 0.022 |
| | **AUC** | 0.714 | 0.500 | 0.501 | 0.501 | 0.501 | 0.501 | 0.502 | 0.502 | 0.502 | 0.502 | 0.502 |
| | **PSNR** | 28.83 | 52.64 | 52.55 | 53.02 | 54.96 | 56.78 | 57.37 | 57.65 | 57.65 | 57.68 | 57.67 |
| **greybox** | **SSIM** | 0.928 | 0.998 | 0.998 | 0.998 | 0.998 | 0.998 | 0.998 | 0.998 | 0.998 | 0.998 | 0.998 |
| | **LPIPS** | 0.028 | 0.001 | 0.001 | 0.001 | 0.001 | 0.001 | 0.001 | 0.001 | 0.001 | 0.001 | 0.001 |
| | **SIFID** | 0.017 | 0.001 | 0.001 | 0.001 | 0.001 | 0.001 | 0.001 | 0.001 | 0.001 | 0.001 | 0.001 |

Table 5: Performance of DwtDctSvd [12] under steganalysis-based removal and the corresponding image quality degradations. NRmv stands for no removal.

| # Images Averaged | | NRmv | 5 | 10 | 20 | 50 | 100 | 200 | 500 | 1000 | 2000 | 5000 |
|---|---|---|---|---|---|---|---|---|---|---|---|---|
| | **Bit Acc** | 1.000 | 0.485 | 0.494 | 0.534 | 0.526 | 0.548 | 0.340 | 0.482 | 0.478 | 0.428 | 0.572 |
| | **PSNR** | 37.59 | 17.89 | 20.58 | 22.28 | 24.67 | 26.88 | 27.74 | 28.62 | 27.41 | 27.46 | 27.57 |
| **Blackbox** | **SSIM** | 0.975 | 0.521 | 0.650 | 0.753 | 0.847 | 0.900 | 0.923 | 0.942 | 0.941 | 0.943 | 0.945 |
| | **LPIPS** | 0.019 | 0.426 | 0.330 | 0.240 | 0.154 | 0.102 | 0.071 | 0.044 | 0.037 | 0.032 | 0.030 |
| | **SIFID** | 0.017 | 1.401 | 0.704 | 0.296 | 0.113 | 0.057 | 0.048 | 0.030 | 0.033 | 0.031 | 0.030 |
| | **Bit Acc** | 1.000 | 0.639 | 0.562 | 0.541 | 0.585 | 0.580 | 0.531 | 0.480 | 0.470 | 0.401 | 0.317 |
| | **PSNR** | 37.59 | 37.74 | 38.21 | 38.38 | 38.47 | 38.61 | 38.67 | 38.70 | 38.71 | 38.69 | 38.67 |
| **greybox** | **SSIM** | 0.975 | 0.967 | 0.972 | 0.973 | 0.974 | 0.976 | 0.976 | 0.976 | 0.976 | 0.977 | 0.977 |
| | **LPIPS** | 0.019 | 0.018 | 0.015 | 0.014 | 0.013 | 0.013 | 0.013 | 0.013 | 0.013 | 0.013 | 0.013 |
| | **SIFID** | 0.017 | 0.012 | 0.010 | 0.010 | 0.009 | 0.009 | 0.009 | 0.009 | 0.009 | 0.009 | 0.009 |

Table 6: Performance (bit accuracy) of RoSteALS [13] under steganalysis-based removal and the corresponding image quality degradations. NRmv stands for no removal.

| # Images Averaged | | NRmv | 5 | 10 | 20 | 50 | 100 | 200 | 500 | 1000 | 2000 | 5000 |
|---|---|---|---|---|---|---|---|---|---|---|---|---|
| | **Bit Acc** | 0.994 | 0.402 | 0.394 | 0.390 | 0.360 | 0.322 | 0.273 | 0.245 | 0.241 | 0.243 | 0.244 |
| | **PSNR** | 28.00 | 17.51 | 19.81 | 21.19 | 22.94 | 24.38 | 24.87 | 25.37 | 24.68 | 24.71 | 24.77 |
| **Blackbox** | **SSIM** | 0.858 | 0.457 | 0.566 | 0.661 | 0.747 | 0.794 | 0.816 | 0.833 | 0.834 | 0.836 | 0.838 |
| | **LPIPS** | 0.039 | 0.395 | 0.305 | 0.224 | 0.149 | 0.109 | 0.087 | 0.068 | 0.065 | 0.061 | 0.059 |
| | **SIFID** | 0.048 | 1.606 | 0.795 | 0.347 | 0.145 | 0.095 | 0.090 | 0.074 | 0.077 | 0.075 | 0.074 |
| | **Bit Acc** | 0.994 | 0.225 | 0.254 | 0.274 | 0.272 | 0.262 | 0.240 | 0.229 | 0.230 | 0.236 | 0.238 |
| | **PSNR** | 28.00 | 26.88 | 27.68 | 28.08 | 28.29 | 28.39 | 28.40 | 28.42 | 28.43 | 28.44 | 28.44 |
| **greybox** | **SSIM** | 0.858 | 0.750 | 0.805 | 0.831 | 0.846 | 0.852 | 0.855 | 0.856 | 0.856 | 0.857 | 0.857 |
| | **LPIPS** | 0.039 | 0.153 | 0.099 | 0.071 | 0.054 | 0.047 | 0.045 | 0.045 | 0.045 | 0.044 | 0.045 |
| | **SIFID** | 0.048 | 0.140 | 0.072 | 0.056 | 0.050 | 0.049 | 0.049 | 0.049 | 0.049 | 0.049 | 0.049 |

Table 7: Performance (bit accuracy) of Gaussian Shading [19] under steganalysis-based removal and the corresponding image quality degradations. NRmv stands for no removal.

| # Images Averaged | | NRmv | 5 | 10 | 20 | 50 | 100 | 200 | 500 | 1000 | 2000 | 5000 |
|---|---|---|---|---|---|---|---|---|---|---|---|---|
| | Bit Acc | 0.999 | 0.490 | 0.469 | 0.537 | 0.488 | 0.486 | 0.479 | 0.461 | 0.463 | 0.465 | 0.462 |
| | PSNR | 9.726 | 9.485 | 9.754 | 9.844 | 9.956 | 10.09 | 10.11 | 10.13 | 10.12 | 10.12 | 10.12 |
| Blackbox | SSIM | 0.322 | 0.164 | 0.197 | 0.242 | 0.268 | 0.286 | 0.294 | 0.301 | 0.305 | 0.306 | 0.307 |
| | LPIPS | 0.613 | 0.686 | 0.672 | 0.648 | 0.636 | 0.631 | 0.632 | 0.632 | 0.632 | 0.631 | 0.632 |
| | SIFID | 0.471 | 0.831 | 0.604 | 0.478 | 0.449 | 0.443 | 0.443 | 0.444 | 0.444 | 0.444 | 0.443 |
| | Bit Acc | 0.999 | 0.507 | 0.501 | 0.540 | 0.490 | 0.490 | 0.479 | 0.462 | 0.461 | 0.462 | 0.462 |
| | PSNR | 9.726 | 9.552 | 9.791 | 9.961 | 10.03 | 10.11 | 10.14 | 10.16 | 10.16 | 10.17 | 10.17 |
| greybox | SSIM | 0.322 | 0.161 | 0.196 | 0.248 | 0.273 | 0.290 | 0.298 | 0.302 | 0.305 | 0.306 | 0.306 |
| | LPIPS | 0.613 | 0.669 | 0.660 | 0.639 | 0.630 | 0.626 | 0.626 | 0.627 | 0.627 | 0.627 | 0.627 |
| | SIFID | 0.471 | 0.802 | 0.597 | 0.474 | 0.449 | 0.444 | 0.439 | 0.437 | 0.438 | 0.438 | 0.438 |

Table 8: Performance (bit accuracy) of Stable Signature [22] under steganalysis-based removal and the corresponding image quality degradations. NRmv stands for no removal.

| # Images Averaged | | NRmv | 5 | 10 | 20 | 50 | 100 | 200 | 500 | 1000 | 2000 | 5000 |
|---|---|---|---|---|---|---|---|---|---|---|---|---|
| | Bit Acc | 0.998 | 0.929 | 0.970 | 0.994 | 0.998 | 0.998 | 0.998 | 0.998 | 0.998 | 0.998 | 0.998 |
| | PSNR | 30.28 | 15.75 | 19.46 | 22.23 | 23.99 | 25.15 | 26.31 | 28.88 | 29.62 | 29.87 | 29.85 |
| Blackbox | SSIM | 0.879 | 0.526 | 0.613 | 0.698 | 0.768 | 0.805 | 0.831 | 0.861 | 0.869 | 0.873 | 0.874 |
| | LPIPS | 0.049 | 0.437 | 0.357 | 0.254 | 0.168 | 0.120 | 0.091 | 0.065 | 0.057 | 0.054 | 0.052 |
| | SIFID | 0.068 | 1.015 | 0.575 | 0.246 | 0.143 | 0.115 | 0.106 | 0.086 | 0.077 | 0.074 | 0.072 |
| | Bit Acc | 0.998 | 0.997 | 0.998 | 0.998 | 0.998 | 0.998 | 0.998 | 0.998 | 0.998 | 0.998 | 0.998 |
| | PSNR | 30.28 | 29.05 | 29.80 | 30.28 | 30.45 | 30.51 | 30.52 | 30.53 | 30.54 | 30.54 | 30.55 |
| greybox | SSIM | 0.879 | 0.790 | 0.833 | 0.860 | 0.871 | 0.875 | 0.876 | 0.877 | 0.878 | 0.878 | 0.878 |
| | LPIPS | 0.049 | 0.167 | 0.115 | 0.074 | 0.056 | 0.052 | 0.050 | 0.050 | 0.049 | 0.049 | 0.049 |
| | SIFID | 0.068 | 0.152 | 0.095 | 0.074 | 0.069 | 0.068 | 0.067 | 0.067 | 0.067 | 0.067 | 0.067 |

Table 9: Performance (bit accuracy) of WmAdapter [27] under steganalysis-based removal and the corresponding image quality degradations. NRmv stands for no removal.

| # Images Averaged | | NRmv | 5 | 10 | 20 | 50 | 100 | 200 | 500 | 1000 | 2000 | 5000 |
|---|---|---|---|---|---|---|---|---|---|---|---|---|
| | Bit Acc | 0.902 | 0.825 | 0.873 | 0.888 | 0.897 | 0.898 | 0.903 | 0.902 | 0.903 | 0.903 | 0.903 |
| | PSNR | 35.18 | 15.74 | 19.67 | 22.75 | 24.80 | 26.17 | 27.74 | 31.78 | 33.13 | 33.60 | 33.60 |
| Blackbox | SSIM | 0.960 | 0.571 | 0.667 | 0.761 | 0.836 | 0.876 | 0.905 | 0.939 | 0.948 | 0.952 | 0.953 |
| | LPIPS | 0.010 | 0.338 | 0.269 | 0.171 | 0.093 | 0.049 | 0.029 | 0.016 | 0.013 | 0.012 | 0.011 |
| | SIFID | 0.004 | 0.318 | 0.155 | 0.050 | 0.023 | 0.016 | 0.015 | 0.010 | 0.007 | 0.006 | 0.006 |
| | Bit Acc | 0.902 | 0.904 | 0.904 | 0.902 | 0.902 | 0.903 | 0.903 | 0.903 | 0.903 | 0.904 | 0.903 |
| | PSNR | 35.18 | 34.12 | 34.70 | 35.04 | 35.18 | 35.22 | 35.23 | 35.25 | 35.26 | 35.26 | 35.26 |
| greybox | SSIM | 0.960 | 0.924 | 0.942 | 0.952 | 0.956 | 0.957 | 0.958 | 0.958 | 0.959 | 0.959 | 0.959 |
| | LPIPS | 0.010 | 0.033 | 0.018 | 0.012 | 0.011 | 0.010 | 0.010 | 0.010 | 0.010 | 0.010 | 0.010 |
| | SIFID | 0.004 | 0.007 | 0.005 | 0.005 | 0.004 | 0.004 | 0.004 | 0.004 | 0.004 | 0.004 | 0.004 |

Table 10: Performance (bit accuracy) of RivaGAN [11] under steganalysis-based removal and the corresponding image quality degradations. NRmv stands for no removal.

| # Images Averaged | | NRmv | 5 | 10 | 20 | 50 | 100 | 200 | 500 | 1000 | 2000 | 5000 |
|---|---|---|---|---|---|---|---|---|---|---|---|---|
| | Bit Acc | 0.973 | 0.738 | 0.800 | 0.864 | 0.902 | 0.930 | 0.946 | 0.959 | 0.961 | 0.963 | 0.967 |
| | PSNR | 39.84 | 17.91 | 20.63 | 22.35 | 24.79 | 27.07 | 27.96 | 28.91 | 27.63 | 27.69 | 27.80 |
| Blackbox | SSIM | 0.979 | 0.525 | 0.654 | 0.757 | 0.851 | 0.904 | 0.925 | 0.945 | 0.944 | 0.946 | 0.948 |
| | LPIPS | 0.036 | 0.426 | 0.331 | 0.243 | 0.159 | 0.110 | 0.083 | 0.059 | 0.054 | 0.050 | 0.049 |
| | SIFID | 0.067 | 1.465 | 0.773 | 0.362 | 0.176 | 0.118 | 0.108 | 0.088 | 0.089 | 0.087 | 0.085 |
| | Bit Acc | 0.973 | 0.972 | 0.973 | 0.972 | 0.972 | 0.972 | 0.972 | 0.972 | 0.972 | 0.973 | 0.973 |
| | PSNR | 39.84 | 39.61 | 40.01 | 40.17 | 40.26 | 40.30 | 40.29 | 40.28 | 40.28 | 40.27 | 40.27 |
| greybox | SSIM | 0.979 | 0.974 | 0.977 | 0.978 | 0.979 | 0.980 | 0.980 | 0.980 | 0.980 | 0.980 | 0.980 |
| | LPIPS | 0.036 | 0.045 | 0.040 | 0.038 | 0.037 | 0.036 | 0.036 | 0.036 | 0.035 | 0.035 | 0.035 |
| | SIFID | 0.067 | 0.085 | 0.074 | 0.070 | 0.067 | 0.066 | 0.065 | 0.065 | 0.065 | 0.065 | 0.065 |

Table 11: Performance (bit accuracy) of SSL [25] under steganalysis-based removal and the corresponding image quality degradations. NRmv stands for no removal.

| # Images Averaged | | NRmv | 5 | 10 | 20 | 50 | 100 | 200 | 500 | 1000 | 2000 | 5000 |
|---|---|---|---|---|---|---|---|---|---|---|---|---|
| **Blackbox** | **Bit Acc** | 0.928 | 0.531 | 0.587 | 0.651 | 0.719 | 0.778 | 0.826 | 0.880 | 0.905 | 0.919 | 0.917 |
| | **PSNR** | 35.05 | 15.78 | 19.77 | 22.85 | 24.88 | 26.22 | 27.81 | 31.80 | 33.07 | 33.52 | 33.52 |
| | **SSIM** | 0.937 | 0.585 | 0.675 | 0.752 | 0.820 | 0.858 | 0.887 | 0.918 | 0.927 | 0.931 | 0.931 |
| | **LPIPS** | 0.040 | 0.302 | 0.234 | 0.155 | 0.098 | 0.069 | 0.055 | 0.046 | 0.043 | 0.042 | 0.041 |
| | **SIFID** | 0.037 | 0.415 | 0.250 | 0.121 | 0.071 | 0.057 | 0.051 | 0.043 | 0.040 | 0.039 | 0.038 |
| **greybox** | **Bit Acc** | 0.928 | 0.895 | 0.912 | 0.921 | 0.924 | 0.923 | 0.926 | 0.927 | 0.929 | 0.929 | 0.929 |
| | **PSNR** | 35.05 | 34.26 | 34.61 | 34.79 | 34.90 | 34.94 | 34.96 | 34.98 | 35.00 | 35.02 | 35.04 |
| | **SSIM** | 0.937 | 0.920 | 0.928 | 0.932 | 0.935 | 0.936 | 0.936 | 0.936 | 0.936 | 0.937 | 0.937 |
| | **LPIPS** | 0.040 | 0.048 | 0.044 | 0.042 | 0.041 | 0.041 | 0.041 | 0.041 | 0.041 | 0.041 | 0.041 |
| | **SIFID** | 0.037 | 0.048 | 0.042 | 0.039 | 0.038 | 0.037 | 0.037 | 0.037 | 0.037 | 0.037 | 0.037 |

Table 12: Performance (bit accuracy) of HiDDeN [10] under steganalysis-based removal and the corresponding image quality degradations. NRmv stands for no removal.

| # Images Averaged | | NRmv | 5 | 10 | 20 | 50 | 100 | 200 | 500 | 1000 | 2000 | 5000 |
|---|---|---|---|---|---|---|---|---|---|---|---|---|
| **Blackbox** | **Bit Acc** | 0.976 | 0.804 | 0.835 | 0.888 | 0.928 | 0.942 | 0.946 | 0.954 | 0.959 | 0.960 | 0.961 |
| | **PSNR** | 36.67 | 17.88 | 20.44 | 21.78 | 24.27 | 26.42 | 28.00 | 28.68 | 27.61 | 27.27 | 27.80 |
| | **SSIM** | 0.956 | 0.521 | 0.639 | 0.738 | 0.839 | 0.886 | 0.913 | 0.931 | 0.931 | 0.932 | 0.936 |
| | **LPIPS** | 0.012 | 0.322 | 0.244 | 0.158 | 0.086 | 0.053 | 0.037 | 0.025 | 0.021 | 0.019 | 0.018 |
| | **SIFID** | 0.018 | 0.643 | 0.348 | 0.156 | 0.058 | 0.037 | 0.030 | 0.026 | 0.025 | 0.024 | 0.023 |
| **greybox** | **Bit Acc** | 0.976 | 0.958 | 0.960 | 0.961 | 0.962 | 0.962 | 0.961 | 0.960 | 0.959 | 0.960 | 0.960 |
| | **PSNR** | 36.67 | 36.00 | 36.45 | 36.69 | 36.85 | 36.90 | 36.93 | 36.97 | 36.99 | 37.01 | 37.04 |
| | **SSIM** | 0.956 | 0.941 | 0.949 | 0.953 | 0.955 | 0.956 | 0.956 | 0.956 | 0.956 | 0.956 | 0.956 |
| | **LPIPS** | 0.012 | 0.017 | 0.014 | 0.013 | 0.012 | 0.012 | 0.012 | 0.012 | 0.012 | 0.012 | 0.012 |
| | **SIFID** | 0.018 | 0.022 | 0.019 | 0.018 | 0.018 | 0.018 | 0.017 | 0.017 | 0.017 | 0.017 | 0.017 |

their own discretion, understanding that it is not a comprehensive solution to the inherent security challenges of content-agnostic watermarking algorithms.

Table 13: Performance (bit accuracy) of DwtDct [12] under steganalysis-based removal and the corresponding image quality degradations. NRmv stands for no removal.

| # Images Averaged | | NRmv | 5 | 10 | 20 | 50 | 100 | 200 | 500 | 1000 | 2000 | 5000 |
|---|---|---|---|---|---|---|---|---|---|---|---|---|
| **Blackbox** | **Bit Acc** | 1.000 | 0.989 | 0.994 | 0.996 | 0.997 | 0.998 | 0.999 | 0.998 | 0.998 | 0.998 | 0.998 |
| | **PSNR** | 37.61 | 17.88 | 20.57 | 22.27 | 24.64 | 26.82 | 27.67 | 28.54 | 27.35 | 27.40 | 27.51 |
| | **SSIM** | 0.962 | 0.518 | 0.645 | 0.745 | 0.836 | 0.887 | 0.909 | 0.927 | 0.926 | 0.928 | 0.930 |
| | **LPIPS** | 0.036 | 0.427 | 0.331 | 0.241 | 0.157 | 0.107 | 0.080 | 0.058 | 0.055 | 0.053 | 0.052 |
| | **SIFID** | 0.063 | 1.454 | 0.766 | 0.352 | 0.168 | 0.110 | 0.096 | 0.079 | 0.080 | 0.078 | 0.076 |
| **greybox** | **Bit Acc** | 1.000 | 1.000 | 1.000 | 1.000 | 1.000 | 1.000 | 1.000 | 1.000 | 1.000 | 1.000 | 1.000 |
| | **PSNR** | 37.61 | 36.79 | 37.17 | 37.31 | 37.39 | 37.46 | 37.49 | 37.49 | 37.49 | 37.47 | 37.46 |
| | **SSIM** | 0.962 | 0.950 | 0.956 | 0.957 | 0.959 | 0.960 | 0.960 | 0.961 | 0.961 | 0.961 | 0.961 |
| | **LPIPS** | 0.036 | 0.042 | 0.039 | 0.038 | 0.037 | 0.037 | 0.036 | 0.036 | 0.036 | 0.036 | 0.036 |
| | **SIFID** | 0.063 | 0.075 | 0.069 | 0.066 | 0.064 | 0.063 | 0.063 | 0.063 | 0.063 | 0.063 | 0.063 |

