# OpenReview forum: "Can Simple Averaging Defeat Modern Watermarks?"
_NeurIPS.cc/2024/Conference — NeurIPS 2024 poster_

### Official Review · Reviewer_VkHo · 2024-07-11

**Soundness:** 3
**Presentation:** 3
**Contribution:** 3
**Rating:** 5
**Confidence:** 4

**Summary:**

This paper introduces a study on the vulnerabilities of digital watermarking techniques to steganalysis attacks. Extensive experiments are conducted to demonstrate the effectiveness of steganalysis in detecting and removing watermarks from images, especially when targeting content-agnostic watermarking methods.

**Strengths:**

1. The paper is well-written and well-organized. The watermark detection and removal/forgery methods are simple and clear.
2. The method of estimating the watermark disturbance by subtracting the average of watermarked images from the average of original images is interesting.
3. The proposed watermark attacks are effective for content-agnostic watermarking methods.

**Weaknesses:**

1. This paper adopts a simple linear assumption that the watermarked image is obtained by adding the content-agnostic perturbation to the original image. The generality of this assumption needs to be clarified, as it forms the theoretical foundation for the watermark attacks discussed in this paper. For instance, the Tree-Ring method embeds information in the frequency domain of initial noise, but why does it conform to this model?

2. The definition of steganalysis needs to be clearly stated, as it is an important concept in this paper.

3. In section 3.1, the paper mentions that ''Instead of applying strong distortions (e.g., noise), the adversary could take a steganalysis approach, first **approximating w** and then crafting minimal distortions to fool D''. However, in the following method, the watermark is never approximated. Instead, the additive perturbation is directly predicted to fool D.

4. In Figure 2, why are the performance metrics inconsistent? Although the caption specifies the use of AUC, the figure mixes AUC with Bit Acc as metrics for different methods.

5. It is better to explain why there is a sudden drop when n = 5 for content-adaptive methods like Stable Signature.

**Questions:**

Please see Weaknesses.

**Limitations:**

The authors have addressed the limitations in section 5.

---

> ### Author Rebuttal · Authors · 2024-08-04
>
> > Why Tree-Ring conforms to the additive Simple Linear Assumption model?
>
> 1. Addition in spatial domain equates to addition in frequency domain: $x(t)+y(t) \xleftrightarrow{\mathcal{F}} X(j\omega)+Y(j\omega)$.
> 2. Tree-Ring's pattern added to the initial noise propagates through the generation pipeline to the final image (Figure 6).
> 3. Thus, Tree-Ring conforms to our model, despite appearing to be a Ring in the frequency domain.
>
>
>
> > The definition of steganalysis needs to be clearly stated, as it is an important concept in this paper.
>
> We appreciate the suggestion. Steganalysis is the detection of hidden data within cover media. In our context, it's applied as a technique for watermark removal. We'll clarify this definition in the revised paper.
>
>
>
> > In section 3.1, the paper mentions that ''Instead of applying strong distortions (e.g., noise), the adversary could take a steganalysis approach, first approximating w and then crafting minimal distortions to fool D''. However, in the following method, the watermark is never approximated. Instead, the additive perturbation is directly predicted to fool D.
>
> Your understanding is correct. It was $\delta_w$ we approximated (denoted as $\hat{\delta}_w$), not $w$. We appreciate this observation and we will revise Section 3.1 to accurately reflect our method.
>
>
>
> > In Figure 2, why are the performance metrics inconsistent? Although the caption specifies the use of AUC, the figure mixes AUC with Bit Acc as metrics for different methods.
>
> Thanks for catching. The caption should be "Performance (AUC or bit accuracy)", not just AUC. We will revise.
>
>
>
> > It is better to explain why there is a sudden drop when n = 5 for content-adaptive methods like Stable Signature.
>
>  - **The point at n=5 isn't a drop, but a discontinuity** between no removal (leftmost point) and watermark removal (other points).
>  - The lowest performance at n=5 occurs because small n values lead to imperfect $\delta_w$ approximations, causing larger image modifications and reduced watermark detection. We're happy to provide further clarification if needed."

---

> > ### Comment · Reviewer_VkHo · 2024-08-11
> >
> > Thanks for your response. Most of my concerns are addressed. I still have questions about additive simple linear assumption model. Specifically, why such pattern added to the initial noise can be propagate through the diffusion pipeline to the final image, as such a generation pipeline is not a simple linear system? While experiments in Figure 6 support this the concept, a theoretical analysis would be beneficial to provide a deeper understanding of the underlying mechanisms.

---

> > > ### Author Response · Authors · 2024-08-12
> > >
> > > To address this question, we provide a brief theoretical analysis that offers insight into the underlying mechanisms.
> > >
> > > Tree-Ring's diffusion pipeline utilizes DDIM sampling. One DDIM denoising step from a noisy image $\mathbf{x}\_t$ to a less noisy one $\mathbf{x}\_{t-\Delta t}$ is described by Equation 13 in [1], which can be rearranged as:
> > >
> > > $$
> > > \mathbf{x}\_{t-\Delta t} = \frac{\sqrt{\alpha\_{t-\Delta t}}}{\sqrt{\alpha\_t}}\mathbf{x}\_t+c\cdot \epsilon\_\theta^{(t)}(\mathbf{x}\_t),
> > > $$
> > >
> > > where $c$ is a time-dependent constant. Tree-Ring can be viewed as adding a systematic bias $\mathbf{\mu}$ (the ripple-like pattern visualized in Figure 6) to an initial noise vector $\mathbf{x}\_T$. Consequently, the first sampling step can be rewritten as:
> > >
> > > $$
> > > \mathbf{x}\_{T-\Delta t} = \frac{\sqrt{\alpha\_{t-\Delta t}}}{\sqrt{\alpha\_t}}(\mathbf{x}\_t + \mathbf{\mu})+c\cdot \epsilon\_\theta^{(t)}(\mathbf{x}\_t + \mathbf{\mu}).
> > > $$
> > >
> > > Empirical evidence suggests that the output of $\epsilon\_\theta$ follows a zero-mean Gaussian distribution, regardless of the timestep and the presence of bias $\mathbf{\mu}$ in the input. Therefore, the term $c\cdot \epsilon\_\theta^{(t)}(\mathbf{x}\_t + \mathbf{\mu})$ does not affect the accumulation of influence from $\frac{\sqrt{\alpha\_{t-\Delta t}}}{\sqrt{\alpha\_t}}\mathbf{\mu}$ through sampling. The accumulated bias term at $\mathbf{x}\_0$ can be expressed as:
> > >
> > > $$
> > > \frac{\sqrt{\alpha\_{t\_1}}}{\sqrt{\alpha\_T}}\cdot
> > > \frac{\sqrt{\alpha\_{t\_2}}}{\sqrt{\alpha\_{t\_1}}}\cdot
> > > \frac{\sqrt{\alpha\_{t\_3}}}{\sqrt{\alpha\_{t\_2}}}\cdots
> > > \frac{\sqrt{\alpha\_{0}}}{\sqrt{\alpha\_1}}
> > > \mathbf{\mathbf{\mu}}
> > > =\frac{\sqrt{\alpha\_0}}{\sqrt{\alpha\_T}}\mathbf{\mu}
> > > =\frac{\sqrt{0.9991}}{\sqrt{0.0047}}\mathbf{\mu}
> > > =14.5799\mathbf{\mu}
> > > $$
> > >
> > > in Tree-Ring's implementation. This calculation demonstrates that the bias $\mathbf{\mu}$ is significantly amplified throughout the generation process.
> > >
> > > Assuming that the low-level pattern is largely preserved through VAE decoding, this amplification explains how a content-agnostic ripple-like pattern can propagate through the complex generation process and manifest in the final images. This analysis provides a theoretical foundation for why the additive simple linear assumption model is still valid with the presence of a diffusion generation pipeline.
> > >
> > > We're happy to help with any more questions or concerns you might have.
> > >
> > >
> > >
> > > [1] Song et al., Denoising Diffusion Implicit Models, ICLR 2021.

---

> > > > ### Comment · Reviewer_VkHo · 2024-08-13
> > > >
> > > > Thanks for your response. I appreciate the authors' efforts. Such attempts are beneficial for explaining this phenomenon, although I personally cannot fully confirm the correctness of the derivation at the moment, especially regarding neglecting the impact of term $\epsilon_{\theta}$.
> > > >
> > > > Considering this method is only for removing content-agnostic watermarks, it is limited by its applicability. I will keep my original rating.

---

> > > > > ### Author Response · Authors · 2024-08-13
> > > > >
> > > > > Thanks for your feedback.
> > > > >
> > > > >
> > > > >
> > > > > **Regarding the impact of $\epsilon_\theta$**:
> > > > >
> > > > > We agree that including the impact of the $\epsilon_\theta$ term could potentially provide deeper insights into the diffusion generation process. However, we respectfully note that a full analysis incorporating the $\epsilon_\theta$ term presents significant challenges due to the high non-linearity of the diffusion denoising network. Such an analysis would likely require extensive additional research and could potentially constitute a separate study in its own right.
> > > > >
> > > > >
> > > > >
> > > > > **Regarding the applicability**:
> > > > >
> > > > > From the perspective of an attack method, this is indeed the limitation, as we have repeatedly addressed in the paper.
> > > > >
> > > > > We believe that, however, the broader impact of our work extends beyond proposing the specific attack technique. The simplicity and effectiveness of stegnanalysis reveals a critical vulnerability in modern watermarking methods, including recent DNN-based approaches. The patterns visualized explains the nature of content-agnostic watermarks, guiding the community towards developing more robust, content-adaptive ones and adopting comprehensive evaluation practices. We believe the contributions have significant potential to advance the field of digital watermarking as a whole.
> > > > >
> > > > >
> > > > >
> > > > > Once again, thank you very much for your insightful feedback.

---

### Official Review · Reviewer_85eL · 2024-07-11

**Soundness:** 2
**Presentation:** 3
**Contribution:** 2
**Rating:** 4
**Confidence:** 3

**Summary:**

The paper introduces steganalysis techniques targeting watermark removal and forgery. The authors demonstrate through experiments that existing content-agnostic watermarking methods are unable to resist steganalysis attacks, advocating for the adoption of content-adaptive strategies.

**Strengths:**

This paper is interesting and well-written. The author introduces steganalysis attack can remove and forge watermarks from watermark images using simple linear addition and subtraction operations in both gray-box and black-box modes.

**Weaknesses:**

1. Although the examples provided by the authors in Figure 7 show that the images after watermark removal exhibit good image quality compared to the watermarked images. However, the PSNR data shown in Figure 2 are not ideal. For instance, in the first example "Tree-Ring," the average PSNR of the removed watermark image is only 17. According to [1],  distortion of modified images with PSNRs lower than 36 dB is noticeable to human visual system. If the quality of the image is severely degraded after watermark removal, then the act of removing the watermark becomes meaningless.
2. The author said “In contrast, traditional distortion-based removal techniques, such as noise perturbations or blurring, typically result in substantial perceptual degradation (as visualized in  Figure 9 in Appendix A.2)”. But I think comparing perceived degradation like this isn't fair because it's difficult to objectively evaluate the severity of different attacks.
3. While the authors present an interesting steganalysis method for removing or forging content-agnostic watermarks, the contribution of this approach may be somewhat limited. Additionally, the method's requirement for a large number of pre-existing watermarked images poses practical challenges for real-world applications.

[1] S. Sarreshtedari and M. A. Akhaee, “A source-channel coding approach to digital image protection and self-recovery,” IEEE Trans. Image Process., vol. 24, no. 7, pp. 2266–2277, Jul. 2015.

**Questions:**

See Weaknesses

**Limitations:**

1. The premise of steganalysis for watermarks requires a large number of images embedded with watermarks, unlike attacks such as JPEG compression, which can directly affect the watermark itself. This limitation makes steganalysis attacks somewhat impractical in real-world applications because attackers may not always have access to a sufficient number of watermarked images.
2. This method is only effective against content-agnostic watermarking.

---

> ### Author Rebuttal · Authors · 2024-08-04
>
> > Why the reported PSNR for Tree-Ring is low (17dB)?
>
> **Short Answer: Our method achieves 29.79dB/34.58dB (blackbox/graybox, n=5000) PSNR when removing Tree-Ring watermarks, indicating minimal visible alteration. The 17dB PSNR value mentioned in the paper measures a different aspect.**
>
> $$
> \text{Clean image } A \overset{\text{Add watermark}}{\longrightarrow} B \overset{\text{Remove watermark}}{\longrightarrow} C
> $$
>
> The 17dB PSNR reported in the paper compares $A$ and $C$, reflecting the cumulative impact of watermarking and removal (Sec 4.1). However, this doesn't represent the degradation caused by watermark removal alone.
>
> To assess removal-induced degradation, we calculate PSNR between $B$ and $C$, yielding 29.79dB/34.58dB (blackbox/graybox, n=5000) for Tree-Ring watermarks.
>
> Tree-Ring, a diffusion-based semantic watermark method, alters image layout significantly. This results in a low PSNR (~16dB) between $A$ and $B$ (Figure 2 NRmv). Consequently, $C$'s PSNR relative to $A$ is also low due to layout differences, despite $C$ closely resembling $B$. This low $A$-$C$ PSNR doesn't accurately reflect the extent of modification during watermark removal.
>
>
>
> > Comparing perceived degradation like this isn't fair because it's difficult to objectively evaluate the severity of different attacks.
>
> **Short Answer: We evaluated both subjectively and objectively. The objective quality evaluation is in Figure 3, which is supplimented by the visualizations (subjective quality evaluation) in your mentioned Figure 9.**
>
> Figure 9 supplements Figure 3 in the main text, which already includes objective quality evaluations. We evaluated four metrics: PSNR, SSIM, LPIPS, and SIFID, and plotted the trade-off curves between watermark removal effectiveness (vertical axis) and image quality degradation (horizontal axis) for different distortion methods. Methods closer to the lower-left corner of the graph are better because they achieve significant watermark removal with minimal distortion. Figure 9 visualizes the data points and their parameters used to plot these curves, thus serving as a complement to Figure 3.
>
>
>
> > The method's requirement for a large number of pre-existing watermarked images poses practical challenges for real-world applications. Attacks such as JPEG compression can directly affect the watermark itself.
>
>  - **The setting is practical:** For example, popular products like Stable Diffusion defaults to adding a fixed watermark to all generated images. Online platforms like Midjourney can add a unique watermark to all images generated by the same user. What's more, these products produce a large number of images everyday. For these popular products, it is possible to obtain tens or hundreds of pre-existing watermarked images.
>
>  - **Steganalysis requires fewer images than recent works:** Many watermark detectors like Tree-Ring have demonstrated robustness against JPEG compression. Recent works have explored adversarial attack-based methods to improve watermark removal. While these methods have successfully removed Tree-Ring watermarks, they require training a surrogate model with 3000 to 7500 images [1, 2]. In comparison, steganalysis requires as few as 500 images for effective removal without significant artifacts, as shown in our experiments.
>
>
>
> > The contribution of the approach may be somewhat limited.
>
> We argue that our work has made significant contributions. As outlined in our common rebuttal:
> - **We are the first to introduce a black-box, training-free method** that effectively removes and forges Tree-Ring watermarks, which is a SOTA diffusion-based watermarking method.
> - **We offer a new perspective on understanding watermarks** by classifying them and visualizing the patterns of content-agnostic watermarks, providing explainability on why these watermarks are vulnerable.
> - **We highlight that addressing the content-agnostic issue is a future direction** for robust watermarking, as we demonstrated the wide impact of simple steganalysis on several modern detectors, such as Tree-Ring, RAWatermark, and RoSteALS.
>
>
>
> **References**
>
> [1] M. Saberi et al., Robustness of AI-image detectors: Fundamental limits and practical attacks, 2023.
>
> [2] B. An et al., Benchmarking the robustness of image watermarks, 2024.

---

> > ### Comment · Reviewer_85eL · 2024-08-13
> >
> > Thank you very much for your detailed response. I appreciate that most of my questions have been addressed. I will increase my score; however, given the inherent limitations of this method,  which is ineffective for content-adaptive issues, I will raise the score to 5 (borderline accept).

---

> > > ### Author Response · Authors · 2024-08-13
> > >
> > > Thank you for addressing our responses positively. We appreciate your recognition of our rebuttal and the contributions of our work. We understand the limitation you've mentioned, which we have repeatedly mentioned in the paper as well.
> > >
> > > Additionally, we kindly wanted to bring to your attention that the score hasn't been updated yet. We appreciate your time and consideration in doing so.

---

> > > > ### Author Response · Authors · 2024-08-14
> > > > **Reminder on Updating the Score**
> > > >
> > > > Just a gentle reminder as the discussion period ends soon -- the score hasn't been updated in the system yet. Thanks again for your consideration!

---

### Official Review · Reviewer_Jv2m · 2024-07-12

**Soundness:** 4
**Presentation:** 3
**Contribution:** 3
**Rating:** 8
**Confidence:** 4

**Summary:**

The paper addresses vulnerabilities in digital watermarking methods, especially those that are content-agnostic. These methods, which embed fixed watermark patterns regardless of image content, are susceptible to steganalysis attacks that can extract and manipulate these patterns, potentially removing or forging watermarks with minimal impact on perceptual quality.

**Strengths:**

The findings of this paper are interesting, and it is reasonably well-written and easy to follow.

1. This paper introduces a novel method for steganalysis that effectively exposes vulnerabilities in content-agnostic digital watermarking techniques. Using simple averaging to extract watermark patterns, the study demonstrates a practical approach that can be applied in gray-box and black-box settings.

2. This paper provides a thorough evaluation of various watermarking methods under different settings. It includes quantitative and qualitative analyses across eight different watermarking techniques, highlighting the particular susceptibility of content-agnostic methods to steganalysis attacks.

3. Recognizing the limitations of current watermarking methods, the paper proposes actionable security guidelines and strategies for enhancing digital watermark security.

**Weaknesses:**

1. The steganalysis method proposed in this paper is fundamentally general and not restricted to images. However, the experiments are confined to analyzing image watermarks. Providing evidence of successful steganalysis in other modalities, such as audio, would further validate and broaden the method's impact.

2. The steganalysis method proposed in this paper can extract a unified low-level watermark pattern from tree-ring watermarked images. So why can the tree-ring change the layout of the generated image?

3. The quality of the figures could be further improved. For example, the font in Figure 2.3.4 is too small. Please enlarge it for easier reading.

**Questions:**

Please see above.

**Limitations:**

Please see above.

---

> ### Author Rebuttal · Authors · 2024-08-04
>
> > The method is fundamentally general and not restricted to images. Providing its effectiveness on other modalities, such as audio, would further validate and broaden the impact.
>
> **You are correct that the issue extends beyond images.** After conducting steganalysis on Audioseal [1] invisible audio watermarks, we found that it also adds content-agnostic watermark that could be removed (<0.76 detection rate). Your suggestion even promotes the significance of our work serving as a reminder of the re-emerging content-agnostic issue.
>
> | Graybox        |   NRmv  |    5    |    10   |    20   |    50   |   100   |   200   |   500   |   1000  |   2000  |   5000  |
> |----------------|:-------:|:-------:|:-------:|:-------:|:-------:|:-------:|:-------:|:-------:|:-------:|:-------:|:-------:|
> | Detection rate | 1.000   | 0.376   | 0.463   | 0.540   | 0.636   | 0.675   | 0.694   | 0.737   | 0.750   | 0.750   | 0.755   |
> | SNR            | 30.432  | 26.647  | 27.435  | 28.690  | 29.516  | 29.828  | 30.107  | 30.102  | 30.305  | 30.408  | 30.343  |
>
> | Blackbox       |   NRmv  |    5    |   10   |   20   |    50   |   100   |   200   |   500   |   1000  |   2000  |   5000  |
> |----------------|:-------:|:-------:|:------:|:------:|:-------:|:-------:|:-------:|:-------:|:-------:|:-------:|:-------:|
> | Detection rate | 1.000   | 0.352   | 0.244  | 0.269  | 0.399   | 0.517   | 0.561   | 0.637   | 0.622   | 0.669   | 0.729   |
> | SNR            | 30.432  | -0.146  | 3.290  | 7.865  | 13.468  | 16.908  | 19.534  | 22.235  | 24.502  | 26.831  | 28.450  |
>
>
>
> > Why can Tree-Ring change the layout of the generated image?
>
> **Adding a Tree-Ring watermark inevitably changes the generation seed.** Replacing the low-frequency part of the initial noise with a Ring pattern will cause global changes of the noise pattern, and hence the layout of the generated image will change. However, we have visualized that Tree-Ring also contains a content-agnostic component, and removing this part of the watermark is enough to fool the tree-ring watermark detector, which makes Tree-Ring vulnerable.
>
>
>
> > The quality of the figures could be further improved.
>
> We will increase the font size in Figure 2-4 to enhance readability, and improve the overall quality of the figures.
>
>
>
> **References**
>
> [1] Robin San Roman et al., Proactive Detection of Voice Cloning with Localized Watermarking, 2024.

---

> ### Comment · Reviewer_Jv2m · 2024-08-10
>
> Thank you for your response. I would appreciate some further clarifications on the following points:
>
> 1. **Regarding the setup of audio experiments**
>
> Could you specify which dataset you used and whether any preprocessing was applied?
>
> 2. **On content-adaptive audio watermarks**
>
> Have you observed any content-adaptive audio watermarks? Are the conclusions consistent with those on images? Could you share some results?
>
> 3. **Clarification on Tree-Ring**
>
> To confirm, are you suggesting that the tree-ring affects the generated images in both high-level aspects (evident in the layout changes of generated content) and low-level aspects (appearing as a content-agnostic pattern overlaid on the generated images)? If this is the case, I would recommend explicitly stating in the paper that "watermarks added in the latent domain of diffusion models may not always result in a semantic watermark and could remain content-agnostic." Highlighting this risk is crucial for future security of diffusion watermarking.

---

> > ### Author Response · Authors · 2024-08-11
> >
> > Thanks for your thoughful feedback. We now respond to your follow-up questions:
> >
> > 1. **Regarding the setup of audio experiments**: We randomly sampled audio clips from the zh-CN subset of the Common Voice dataset [1]. For black-box settings, we ensured that the unwatermarked and watermarked audio sets were disjoint. All audio clips were sampled at 16kHz and preprocessed by cropping to retain only the first two seconds. This setup ensures that all audio clips contain watermarked segments and are aligned [2, 3].
> >
> > 2. **On content-adaptive audio watermarks**: Yes, we observed that WavMark [2] adds content-adaptive watermarks to audio clips.
> >
> > | Blackbox |   NRmv  |    5    |    10   |   20   |   50   |   100  |   200   |   500   |   1000  |   2000  |   5000  |
> > |----------|:-------:|:-------:|:-------:|:------:|:------:|:------:|:-------:|:-------:|:-------:|:-------:|:-------:|
> > | Bit Acc  | 0.82    | 0.0088  | 0       | 0      | 0.01   | 0.0256 | 0.0294  | 0.1062  | 0.25    | 0.3669  | 0.85    |
> > | SNR (dB) | 44.0328 | -6.7481 | -4.0659 | 0.0338 | 5.5133 | 9.1686 | 13.0371 | 17.1913 | 20.6909 | 24.5124 | 28.1818 |
> >
> > When averaging 5000 audio samples, the bit accuracy after watermark removal still reached 0.85. This demonstrates that steganalysis is not effective in reducing the watermark's bit accuracy. This finding is consistent with our conclusions from image experiments, indicating that across various modalities, our method can remove content-agnostic watermarks but is ineffective against content-adaptive watermarks.
> >
> > 3. **Clarification on Tree-Ring**: Exactly. We are demonstrating that adding a Tree-Ring watermark affects both aspects. Tree-Ring inevitably changes the initial noise, which leads to changes in layout. This has been visualized in the Tree-Ring paper. Simultaneously, our experiments prove that Tree-Ring's watermarking process also adds a content-agnostic component. It is primarily this content-agnostic pattern that affects Tree-Ring's watermark detection.
> >
> > ---
> >
> > [1] R. Ardila et al., Common voice: a massively-multilingual speech corpus, LREC 2020.
> >
> > [2] G. Chen, WavMark: Watermarking for audio generation, Arxiv 2023.
> >
> > [3] R. S. Roman, Proactie detection of voice cloning with localized watermarking, ICML 2024.

---

> ### Comment · Reviewer_Jv2m · 2024-08-13
>
> Thank you for the authors' clarification and extra experiments. My concerns have been addressed. I will raise my score to 8. If possible, please incorporate the extra experiments and discussion into the next version of the paper.

---

> > ### Author Response · Authors · 2024-08-14
> >
> > Thanks for your thoughtful feedback and for raising the score. We'll include the additional experiments and discussion, which we also believe could broaden the impact of our work.

---

### Official Review · Reviewer_aL14 · 2024-07-23

**Soundness:** 3
**Presentation:** 3
**Contribution:** 1
**Rating:** 5
**Confidence:** 3

**Summary:**

This paper introduces a steganalysis-based attack aimed at image watermarking methods. The attack is effective in both gray-box and black-box scenarios and focuses on identifying repeating patterns present in watermarked images. These patterns can be exploited to either remove watermarks or add them to non-watermarked images without needing access to the original watermarking algorithm. Experimental results demonstrate that this attack can successfully compromise watermarks that use content-agnostic watermark patterns.

**Strengths:**

- The authors illustrate the vulnerability of content-agnostic watermarks to simple attacks, serving as a good reminder for researchers to ensure their watermarks are content-aware.

- The proposed attack is simple and compared to some existing watermark attacks, has a lower computational cost. However, it's worth noting that the computational cost of other attacks is usually not significant either, so the value of this simplicity and speed-up remains debatable.

**Weaknesses:**

- Similar ideas (i.e., considering watermarked and non-watermarked samples to extract watermark patterns for removal and forgery) have been explored in existing works [1][2]. For instance, [1] discusses a spoofing attack using a noise image $X$, watermarking it to obtain $X_{wm}$, and then adding $(X_{wm}-X)$ to clean images to create watermarked versions. Both [1] and [2] also describe a surrogate model adversarial attack, which involves training a CNN to distinguish between watermarked and non-watermarked images. This CNN is then used to adversarially alter samples to change their watermark status. This adversarial attack is a more advanced version of the method suggested in this paper and can potentially target content-aware watermarks as well.

- The authors claim, "Notably, our steganalysis-based approach is the first to effectively remove Tree-Ring watermarks without access to the algorithm," which is inaccurate. The adversarial attack proposed in [1], though ML-based, requires only a set of watermarked and non-watermarked samples (similar to the black-box attack described in this paper) to break Tree-Ring.

- The method introduced in this paper lacks novelty, and the vulnerability of content-agnostic watermarks to this type of attack is trivial and not a significant discovery.

[1] Saberi, M., Sadasivan, V. S., Rezaei, K., Kumar, A., Chegini, A., Wang, W., and Feizi, S. Robustness of AI-image detectors: Fundamental limits and practical attacks, 2023.

[2] An, B., Ding, M., Rabbani, T., Agrawal, A., Xu, Y., Deng, C., Zhu, S., Mohamed, A., Wen, Y., Goldstein, T., et al.: Benchmarking the robustness of image watermarks, 2024.

**Questions:**

- I suggest that the authors compare their attack to existing watermark attacks (e.g., from this watermark benchmark [1]). This comparison is necessary to illustrate the advantages of their attacks.

- The results for the forgery attack are provided only for the Tree-Ring watermark. I recommend including results for other watermarks as well to provide a more comprehensive evaluation.


[1] An, B., Ding, M., Rabbani, T., Agrawal, A., Xu, Y., Deng, C., Zhu, S., Mohamed, A., Wen, Y., Goldstein, T., et al.: Benchmarking the robustness of image watermarks, 2024.

**Limitations:**

- The attack only works on content-agnostic watermarks, which the authors adequately mentioned in several parts of the paper.

---

> ### Author Rebuttal · Authors · 2024-08-04
>
> > Similar ideas have been explored in existing works [1, 2].
>
> We argue that our work fundamentally differs from [1, 2] with distinctive advantages.
>
> **Differences with the spoofing attack in [1]**
>
> The spoofing attack described in [1] requires access to the watermark encoder (white-box) and does not generalize to removal, while our steganalysis-based approach unifies removal and forgery without accessing the decoder (black-box). In terms of explainability, our method extends beyond [1] by using steganalysis to extract content-agnostic watermarks. This step enables visualizing to show the existence of the extracted and added patterns during forgery, whereas [1], lacking this step, cannot explain why and which component of the weak overlayed watermark signals enables spoofing. We thus believe the idea explored by [1] is fundamentally different from our work.
>
> **Differences with adversarial attacks in [1, 2]**
>
> Steganalysis, adversarial, and distortion-based attacks are parallel methods, each fundamentally different from the others. Steganalysis attacks involve extracting hidden information to facilitate removal, adversarial attacks focus on solving optimization problems to harness adversarial examples, and distortion-based attacks typically employ image-processing techniques to create distortions. They are not variations of one another; therefore, we don't think one is a more advanced version of another.
>
> Specifically, steganalysis has advantages over the adversarial attacks described in [1, 2]: our approach can effectively remove watermarks from just 500 images without leaving noticeable artifacts, seamlessly unifying both removal and forgery under whitebox and blackbox scenarios. In contrast, the adversarial attacks in [1, 2] require at least 3000 images for training. The attack in [2] also does not support forgery, as perturbations disrupt the entire latent space. The claim about the potential applicability of AdvCls to content-aware watermarks remains debatable [1, 2], as both work only demonstrated effectiveness in removing Tree-Ring watermarks unless with large perturbations ($\epsilon>10$) [1]. [1] fails on StegaStamp. [2] fails on Stable Signature and StegaStamp.
>
> **In conclusion**, the uniqueness of steganalysis seamlessly unifies removal and forgery with visualizable explainability under both whitebox and blackbox settings. It requires significantly fewer images than adversarial attacks, further showcasing its advantage over existing approaches.
>
>
>
> > Steganalysis is not the first blackbox method, as [1] also works under blackbox.
>
> Thanks for pointing out. We will rephrase to clarify that our method is the first that is training-free.
>
>
>
> > The method lacks novelty, and the vulnerability is trivial.
>
> We argue that our method is novel, and the vulnerability is severe. As outlined in our common rebuttal:
> - **Novelty**
>   - We are the first to introduce a blackbox training-free method that effectively removes and forges Tree-Ring watermarks.
>   - We are the first to explain what modern content-agnostic watermark patterns look like (such as the Tree-Ring ripples), and why they can be easily removed.
>   - We highlight a future direction for robust watermarking: to address the content-agnostic issue by evaluating against steganalysis attacks, similar to how robustness is evaluated against distortions.
> - **Severity and Impact**: Our experiments show that a sufficiently simple and straightforward method easily fools several modern watermark detectors. The vulnerability is therefore severe with wide impact.
>
>
>
> > Comparison with the watermark benchmark [2].
>
> We compared with attacks in [2] following its Detection setting, reporting TPR@0.1%FPR (`low_1000` in its code). The comparison is on Tree-Ring watermarks, which is the watermark that could be successfully removed by both [2] and our work.
>
> |Method|TPR@0.1%FPR|
> |---|---|
> |Dist-Rotation / Rcrop / Erase / Bright / Contrast / Blur / Noise / JPEG|0.009 / 0.013 / 1.000 / 0.992 / 0.995 / 0.230 / 0.945 / 0.688|
> |DistCom-Geo / Photo / Deg|0.016 / 0.994 / 0.726|
> |Regen-Diff / DiffP / VAE / KLVAE / 2xDiff / 4xDiff|0.407 / 0.397 / 0.464 / 0.964 / 0.222 / 0.171|
> |AdvEmbG-KLVAE8 / RN18 / CLIP / KLVAE16 / SdxlVAE|0.152 / 0.898 / 0.853 / 0.751 / 0.835|
> |AdvCls-UnWM&WM / Real&WM / WM1&WM2|0.291 / 1.000 / 0.253|
> |Ours n=5 / 10 / 20 / 50 / 100 / 200 / 500 / 1000 / 2000 / 5000|**0.000** / **0.000** / **0.000** / **0.000** / **0.000** / **0.000** / **0.000** / **0.000** / **0.000** / **0.000**|
>
> As can be seen, our removal lowers TPR@0.1%FPR to near 0, more effective than any attacks in [2].
>
>
>
> > Include forgery results for other watermarks to provide a more comprehensive evaluation.
>
> |Method/Metric|Setting|NoForgery|5|10|20|50|100|200|500|1000|2000|5000|
> |---|---|:---:|:---:|:---:|:---:|:---:|:---:|:---:|:---:|:---:|:---:|:---:|
> |ROS (Bit Acc)|Graybox|0.506|0.990|0.988|0.987|0.987|0.987|0.988|0.989|0.988|0.988|0.988|
> |RAW (AUC)|Graybox|0.500|0.732|0.732|0.730|0.727|0.723|0.722|0.721|0.721|0.721|0.721|
> |SVD (Bit Acc)|Graybox|0.520|0.568|0.558|0.554|0.562|0.563|0.560|0.561|0.559|0.562|0.557|
> |ROS (Bit Acc)|Blackbox|0.506|0.900|0.926|0.953|0.968|0.978|0.982|0.984|0.983|0.983|0.982|
> |RAW (AUC)|Blackbox|0.500|0.010|0.018|0.038|0.051|0.069|0.085|0.127|0.184|0.222|0.243|
> |SVD (Bit Acc)|Blackbox|0.520|0.530|0.530|0.510|0.528|0.535|0.480|0.517|0.488|0.493|0.517|
>
> - **RoSteALS (ROS)**: highly effective (0.9+ bit acc), suggesting RoSteALS adds linearly content-agnostic watermarks
> - **RAWatermark (RAW)**: moderately effective under graybox (0.72 AUC) but behave like removal under blackbox (0.25 AUC)
> - **DwtDctSvd (SVD)**: ineffective forgery, suggesting the effective removal could be due to $\delta_w$ disrupting the decoding space (similar to AdvCls-WM1&WM2 in [2])
>
>
>
> **References**
>
> [1] M. Saberi et al., Robustness of AI-image detectors: Fundamental limits and practical attacks, 2023.
>
> [2] B. An et al., Benchmarking the robustness of image watermarks, 2024.

---

> ### Comment · Reviewer_aL14 · 2024-08-10
>
> > The spoofing attack described in [1] requires access to the watermark encoder (white-box) and does not generalize to removal, while our steganalysis-based approach unifies removal and forgery without accessing the decoder (black-box).
>
> I understand the point of authors about their attack being black-box, in contrast to the spoofing attack from [1]. However, the spoofing attack is very similar to the steganalysis attack from this paper, and while [1] does not discuss on generalization of the attack on watermark removal, I don't see a reason why it wouldn't work.
>
> > It requires significantly fewer images than adversarial attacks, further showcasing its advantage over existing approaches.
>
> Reading the authors' response, I accept that their method, while having similarities to the adversarial attack, has its own advantages and disadvantages (only working on content-unaware watermarks).
>
> I still believe that the vulnerability of content-unaware watermarks to this attack is trivial (especially in case of the gray-box setting). However, for the black-box setting, where non-paired samples for watermarked and non-watermarked images are used, there is more novelty and contribution. I appreciate the authors for adding new results for comparison to the attacks from [2], and the forgery results. I will increase my score to 5 (borderline accept).
>
>
>
>
>
>
>
> [1] M. Saberi et al., Robustness of AI-image detectors: Fundamental limits and practical attacks, 2023.
>
> [2] B. An et al., Benchmarking the robustness of image watermarks, 2024.

---

> > ### Author Response · Authors · 2024-08-12
> >
> > Thank you for your follow-up comments and recognizing our work's strengths and contributions. We appreciate the increased score.
> >
> >
> >
> > Regarding your concerns:
> >
> >
> >
> > > The spoofing attack is very similar to the steganalysis attack from this paper, and while [1] does not discuss on generalization of the attack on watermark removal, I don't see a reason why it wouldn't work.
> >
> > [1]'s spoofing method requires **(1)** access to the model with **(2)** the original watermark key (bit sequence/Tree-Ring pattern), which limits it to whitebox attacks. In contrast, our method is capable of performing blackbox attacks, as it extends beyond [1] by utilizing steganalysis to extract content-agnostic watermarks. This also offers better explainability by enabling the visualization of watermark patterns extracted.
> >
> > We generalized [1] to watermark removal with the following setup:
> >
> > $$
> >     \text{Original spoofing: \quad}  x_{\text{spoofed}} = \beta x_{\text{clean}} + (1-\beta) w, \quad \beta = 0.7;
> > $$
> > $$
> >     \text{Generalized removal: \quad}  x_{\text{removed}} = \beta x_{\text{w}} + (1-\beta) w, \quad \beta = 1.3.
> > $$
> >
> > Our black-box method reduced Tree-Ring's AUC to 0.241 with a PSNR of 29.79dB (before and after removal), while [1]'s removal only achieved 0.929 AUC with 22.71dB PSNR. **This demonstrates that [1] is ineffective at removing Tree-Ring watermarks, whereas our method achieves superior removal with less quality degradation.**
> >
> >
> >
> > > I still believe that the vulnerability of content-unaware watermarks to this attack is trivial (especially in case of the gray-box setting).
> >
> > 1. We're the first to identify and systematically evaluate the content-agnostic vulnerability in modern watermarking systems.
> > 2. Our finding, though simple, has broad implications for various watermarking methods. Highlighting this blind spot is crucial for guiding future research towards steganalysis-secure approaches.
> > 3. Our graybox setting provides explainability to the content-agnostic issue. It enriches the application scenario and serves as a complement of the attack system.
> >
> >
> >
> > [1] M. Saberi et al., Robustness of AI-image detectors: Fundamental limits and practical attacks, 2023.

---

> > > ### Author Response · Authors · 2024-08-12
> > >
> > > Please feel free to raise any further concerns or questions. We would be glad to provide additional details.

---

### Author Rebuttal · Authors · 2024-08-05

### Common Response

We appreciate the reviewers' comments and the opportunity for rebuttal. Here we would like to clarify the significance and contributions of our work.

&nbsp;

**Our contributions**
- **We introduce the first blackbox, training-free method** that successfully removed and forged Tree-Ring watermarks. This method integrates watermark removal and forgery into a single operation, enhancing diversity in watermark analysis.
- **We offer a new perspective on understanding watermarks**: By classifying watermarks into content-agnostic and content-adaptive, we provide explainability. We visualize the extracted content-agnostic patterns (such as the ripple-like pattern in Tree-Ring or the vertical bars in DwtDctSvd's $C_B$ chroma). This perspective helps explain why some watermarks are vulnerable to steganalysis.
- **We highlight a future direction for robust watermarking: addressing the content-agnostic issue.** We identify the recurring problem where recent works, despite their methodological complexity, still add fixed patterns without considering the image content. For example, we visualized that a Tree-Ring pattern could propagete through a DDIM diffusion sampling process still being content-agnostic, and watermark decoding relies on this pattern. This allows a sufficiently simple steganalysis to easily deceive several modern watermark detectors, including DNN-involved methods like Tree-Ring, RAWatermark, and RoSteALS. We provide security guidelines, recommending that future researchers include additional evaluations against simple steganalysis attacks, contributing to the security and advancement of watermarking community.

---

> ### Author Response · Authors · 2024-08-14
> **Thank you for engaging in the discussions**
>
> Dear Reviewers,
>
> Thank you all for engaging so thoughtfully with our work during the discussion period. We greatly appreciate your constructive feedback. We're also grateful that most of you considered raising the scores. We wish you all the best in your research and future career!
>
> Best regards,
> #9663 authors

---

### Decision · Program_Chairs · 2024-09-25

**Decision:**

Accept (poster)

**Comment:**

The real average rating of this paper should be 5.75 since reviewer 85eL said that the score should be bumped up to 5 but still remained as 4 (no response after emailing).

The paper introduces a steganalysis-based attack targeting content-agnostic image watermarking methods. Reviewers acknowledge that the paper effectively demonstrates vulnerabilities in these watermarking techniques (aL14, Jv2m, VkHo). However, concerns are raised about the novelty of the method, as similar ideas have been explored in prior works (aL14). Questions arise regarding the generality of the assumptions made, particularly for methods like Tree-Ring that embed watermarks in the frequency domain (Jv2m, VkHo). Practical impact is questioned due to potential image quality degradation after watermark removal (85eL) and the requirement for a large number of pre-existing watermarked images (85eL, aL14). While the paper is well-written and organized (Jv2m, 85eL, VkHo), the contribution is considered limited due to existing similar research and practical challenges in real-world applications.

In the rebuttal the authors address concerns about the novelty of their steganalysis-based attack on content-agnostic watermarks. They emphasize that their method is the first black-box, training-free approach that unifies both watermark removal and forgery without needing access to the watermark encoder or decoder, differing fundamentally from prior works. Additional experiments were provided comparing their method with existing attacks, demonstrating effectiveness with fewer images and minimal perceptual degradation. They also clarify that the vulnerability is severe and not trivial, extending their findings to other modalities like audio. The AC recommends accept as most of the reviewers acknowledged their concerns were addressed and bumped up their ratings.